# Targeted inhibition of ubiquitin signaling reverses metabolic reprogramming and suppresses glioblastoma growth

Rossella Delle Donne [1,10], Rosa Iannucci [1,10], Laura Rinaldi[1], Luca Roberto[2], Maria A. Oliva[3], Emanuela Senatore[1], Domenica Borzacchiello[1], Luca Lignitto[1], Giorgio Giurato[4], Francesca Rizzo [4], Assunta Sellitto[4], Francesco Chiuso[1], Salvatore Castaldo[3], Giovanni Scala[5], Virginia Campani[6], Valeria Nele[6], Giuseppe De Rosa[6], Chiara D'Ambrosio [7], Corrado Garbi[1], Andrea Scaloni[7], Alessandro Weisz[4,8], Concetta Ambrosino[2,9], Antonella Arcella[3] & Antonio Feliciello [1✉]

Glioblastoma multiforme (GBM) is the most frequent and aggressive form of primary brain tumor in the adult population; its high recurrence rate and resistance to current therapeutics urgently demand a better therapy. Regulation of protein stability by the ubiquitin proteasome system (UPS) represents an important control mechanism of cell growth. UPS deregulation is mechanistically linked to the development and progression of a variety of human cancers, including GBM. Thus, the UPS represents a potentially valuable target for GBM treatment. Using an integrated approach that includes proteomics, transcriptomics and metabolic pro-filing, we identify praja2, a RING E3 ubiquitin ligase, as the key component of a signaling network that regulates GBM cell growth and metabolism. Praja2 is preferentially expressed in primary GBM lesions expressing the wild-type isocitrate dehydrogenase 1 gene (IDH1). Mechanistically, we found that praja2 ubiquitylates and degrades the kinase suppressor of Ras 2 (KSR2). As a consequence, praja2 restrains the activity of downstream AMP-dependent protein kinase in GBM cells and attenuates the oxidative metabolism. Delivery in the brain of siRNA targeting praja2 by transferrin-targeted self-assembling nanoparticles (SANPs) prevented KSR2 degradation and inhibited GBM growth, reducing the size of the tumor and prolonging the survival rate of treated mice. These data identify praja2 as an essential regulator of cancer cell metabolism, and as a potential therapeutic target to suppress GBM growth.

[1] Department of Molecular Medicine and Medical Biotechnology, University Federico II, Naples, Italy. [2] Biogem, Ariano Irpino, Avellino, Italy. [3] I.R.C.C.S Neuromed, Pozzilli (Isernia), Italy. [4] Laboratory of Molecular Medicine and Genomics, Department of Medicine, Surgery and Dentistry SMS, University of Salerno, Salerno, Italy. [5] Department of Biology, University Federico II, Naples, Italy. [6] Department of Pharmacy, University Federico II, Naples, Italy. [7] Proteomics, Metabolomics and Mass Spectrometry Laboratory, ISPAAM, National Research Council, Portici (Naples), Italy. [8] Genome Research Center for Health, Campus of Medicine, University of Salerno, Salerno, Italy. [9] Department of Science and Technology University of Sannio, Benevento, Italy. [10]These authors contributed equally: Rossella Delle Donne, Rosa Iannucci. ✉email: feliciel@unina.it

Glioblastoma (GBM) represents 12–15% of all intracranial neoplasms and 60–75% of glial tumors[1]. The incidence of GBM is 3–4 cases per 100,000 individuals, with a peak at age 45–75 years and a prevalence in males. GBM is characterized by high neo-angiogenesis, pronounced mitotic activity, cellular heterogeneity, high proliferative rate, and necrosis. In addition, the presence of tumor stem cells, able to proliferate and generate glial neoplastic cells[2,3], contributes to the poor prognosis of patients with GBM, whose average survival is about 12 months from diagnosis. Based on the histologic characteristics and the presence of nuclear atypia, mitosis, and endothelial proliferation, glioma lesions have been divided into four subgroups: grade I (lesions with low proliferative potential), grade II (lesions with low proliferative potential, but with a tendency to infiltration and with cytological atypia), grade III (lesions with evidence of anaplasia and mitotic activity) and grade IV (lesions with nuclear atypia, cell pleomorphism, mitotic activity, microvascular proliferation and/or necrosis)[1]. The morphological classification of gliomas has been recently integrated with molecular and predictive data that are very important for the prognosis and the limited therapeutic options. In particular, the presence of mutations of the isocitrate dehydrogenase 1 gene (IDH1) and co-deletion of chromosomes 1p–19q have become determining factors in the definition of different histo-molecular subtypes. Therefore, tumors with the simultaneous presence of IDH1 gene mutation and 1p–19q co-deletion are classified as oligodendroglial forms, while tumors with IDH1 gene mutation and absence of 1p–19q co-deletion can be assigned to astrocytic forms, as well as tumors without IDH1 gene mutation[3–6]. Although efforts have been made to identify molecular pathways and potential therapeutic targets involved in gliomagenesis, total tumor resection followed by adjuvant chemotherapy and radiotherapy remains the standard of care[7–9]. However, in most cases, therapy is ineffective to control the tumor growth, which often recurs after a period of time that varies from patient to patient. Furthermore, the intratumoral heterogeneity of GBM complicates the clinical outcome of the therapy, demanding the discovery of novel therapeutic targets for this disease[10,11].

The ubiquitin-proteasome system (UPS) emerged as an important component of the cellular machinery controlling protein fate and activity. By modulating the ubiquitylation state of a target protein, UPS regulates key biological functions, including growth, metabolism, differentiation, and development[12]. UPS controls different steps of tumor development, progression, and spreading. In GBM cells, regulators and effectors of UPS have been causally implicated in key tumor cell functions, such as the activation of membrane receptors and mitogenic signaling, cell survival pathways, DNA damage repair, regulation of gene transcription, and stem cell maintenance[13]. Thus, UPS represents a potential source for novel therapeutic strategies for GBM treatment[14]. We have identified a RING E3 ubiquitin ligase, praja2, widely expressed in mammalian cells and tissues which is involved in essential aspects of cell physiology. Several signaling pathways are regulated by praja2. The stability of regulatory PKA subunits is directly regulated by praja2, which binds and tethers cAMP-dependent protein kinase (PKA) to intracellular membranes and organelles, ensuring efficient integration, propagation, and amplification of the locally-generated cAMP to distinct target compartments[15–17]. By regulating cAMP signaling, praja2 efficiently couples phosphorylation to ubiquitination of protein kinases, scaffolds, and effectors, with important implications for neuronal activity, development, inflammatory responses, ciliogenesis, cell growth, and metabolism[18–28]. Dysregulation of praja2-regulated signaling pathways has been causally linked to the growth and progression of GBM[29,30]. In GBM cells, praja2 ubiquitylates and degrades MOB1, which is the regulatory subunit of LATS1/2 kinase and the positive regulator of the oncosuppressive Hippo pathway. Proteolysis of MOB1 modulated by praja2 stimulated the growth of GBM[29]. The negative regulation of the Hippo pathway by praja2 is pathogenically relevant also for renal fibrosis, a final common pathological feature of chronic kidney disease (CKD), which is characterized by tubular atrophy, interstitial fibrosis, and glomerulosclerosis. In CKD, the interaction between praja2 and MOB1 is enhanced by kindlin-2, a FERM-containing focal adhesion protein that is abundantly expressed in mesodermal tissues. Inhibition of kindlin-2 prevents praja2-mediated proteolysis of MOB1 and alleviates the renal fibrotic phenotype[31]. We have hypothesized that the ultimate biological effect induced by praja2 is a metabolic switch which drives development and cancer growth. The GBM is an important model because the role of this ligase and UPS in the control of metabolic pathways in GBM cells was, so far, largely unknown.

We report, here, the essential role of praja2 and UPS in the regulatory networks underlying the metabolic reprogramming of GBM cells. We have identified a novel target of praja2 that is relevant for tumor growth: the kinase suppressor of Ras 2 (KSR2). We used a strategy based on transferrin-targeted self-assembling nanoparticles (SANPs) to deliver inhibitory RNA molecules targeting praja2 to the brain, thus demonstrating that praja2 is a relevant therapeutic target for GBM treatment and suggesting novel therapies based on RNA delivery.

## Results

**Praja2 ubiquitylates and degrades KSR2.** To dissect the biological role of praja2 in GBM, we first analyzed the expression profile of praja2 in 20 human glioma biopsies of patients undergoing brain surgery, which were previously characterized for the presence/absence of IDH1 mutations. Immunostaining analysis revealed high levels of praja2 in primary GBM tissues carrying wild-type IDH1, compared to mutant IDH1 low-grade glioma (astrocytoma) and secondary GBM (Fig. 1a, b). We also evaluated praja2 mRNA expression profile in a TCGA-GBM cohort. Samples were then stratified based on their IDH status (wild-type vs. mutant) and histological subtype (astrocytoma, glioblastoma, oligoastrocytoma, oligodendroglioma). The analysis was performed on 167 samples of astrocytoma (51 wild-types and 116 mutant), 235 samples of glioblastoma (216 wild-type and 19 mutant), 113 samples of oligoastrocytoma (15 wild-types and 98 mutant), and 166 samples of oligodendroglioma (16 wild-type and 150 mutant). We found that praja2 was expressed at higher levels in IDH1 wild-type astrocytoma, oligoastrocytoma, and glioblastoma, compared to the counterpart glioma lesions carrying mutant IDH1 (Supplementary Fig. 1a).

To delineate the mechanisms of praja2 action and identify relevant partners and regulators of cancer cell growth and metabolism, we performed proteomic analysis of affinity-purified praja2 complexes from total cell lysates. HEK293 cells overexpressing Flag-praja2rm and Flag were comparatively evaluated, as described in Methods. The praja2 inactive mutant praja2rm has no ligase activity, while it still binds substrates/partners[15,18,19,21,29]. Proteomic analysis of praja2 complexes selectively identified a variety of gene products involved in different metabolic pathways. Praja2 interactors identified in these experiments and those identified by other groups, reported in Intact, STRING and CPTAC-GBM protein databases, were used as input to generate a protein–protein interaction (PPI) network (Fig. 2a, Supplementary Fig. 1b, and Supplementary Data 1). Network analysis identified the 5' AMP-activated protein kinase (AMPKα1 and AMPKγ1), a metabolic sensor and regulator of energy homeostatic processes, as putative praja2-interacting partner[32]. Although KSR2, and its homolog KSR1,

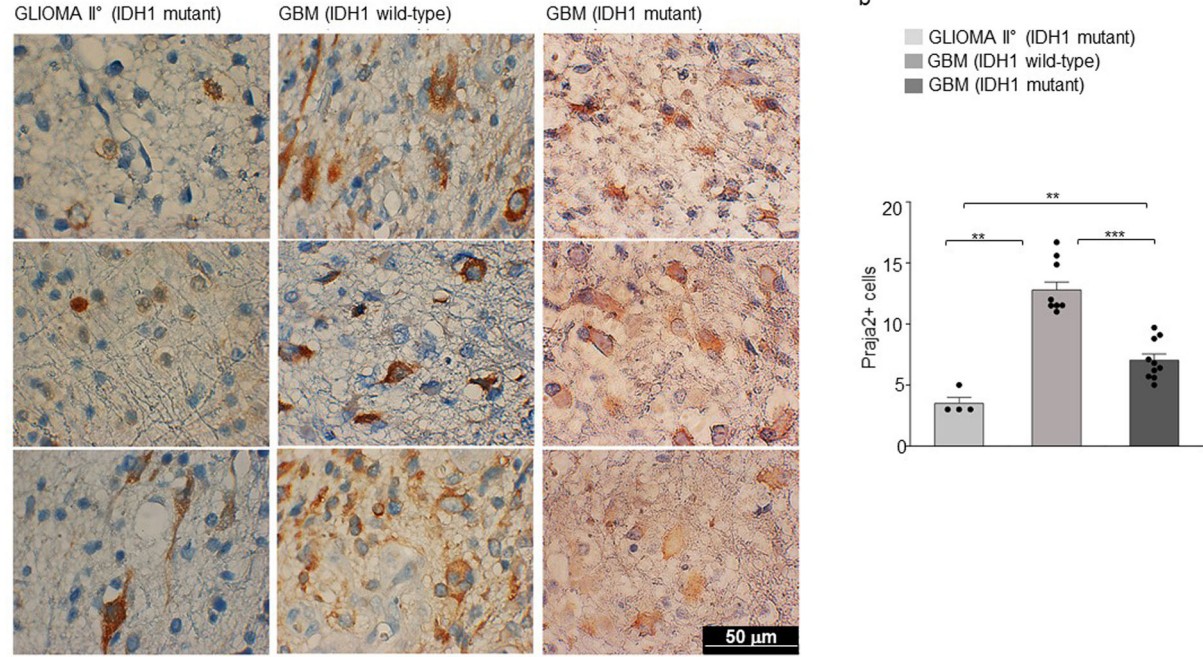

**Fig. 1 Expression analysis of praja2 in glioma tissues. a** Tissue sections from human astrocytoma grade II and GBM lesions were immunostained with anti-praja2 antibody and analyzed by stereological microscope. The images are representative of praja2 staining in low-grade (astrocytoma II) and high-grade (GBM) gliomas. IDH1 gene mutation status in tumor lesions is indicated. Images were acquired at 100X magnifications. Scale bar = 50 μm. **b** Graph represents the levels of praja2 in different histological types of glioma and are expressed as the density of positive cells for microscope fields. *P* value: ** = 0.002; ** = 0.007; *** = 0.00015. The IHC analysis was carried out on a total of 20 gliomas (ten with wild-type IDH1 and ten gliomas with IDH1-R132 mutation).

were originally identified as regulators of the Ras-Raf-MEK-dependent mitogenic pathway, KSR2 has a prominent role in energy intake and metabolism[32–36]. As a consequence of increased metabolic demands, KSR2 activates AMPK which, in turn, promotes glucose and fatty acid oxidation. Variants of KSR2 identified in obese children affect the Ras-Raf pathway and oxidative metabolism, causing a reduced heart rate, low basal metabolism, and severe insulin resistance[37]. Based on these observations, we defined the role of praja2 in the KSR2-AMPK metabolic pathway. First, we confirmed by co-immunoprecipitation that praja2 and KSR2 form a stable complex in cell lysates (Fig. 2b). Deletion mutagenesis and binding analysis identified the domain of praja2 located at residues 530–630 as essential for the interaction with KSR2 (Fig. 2b). GST pulldown experiments confirmed that praja2 directly interacts with KSR2 (Fig. 2c).

Immunofluorescence analysis revealed that a portion of praja2 and KSR2 colocalizes in the cytoplasm of GBM cells (Pearson's coefficient value of ~0.6 Fig. 2d). Co-immunoprecipitation assays demonstrated that both KSR2 and AMPKα1 form a complex with praja2 (Fig. 2e and Supplementary Fig. 2a). Since praja2 acts as an E3 ubiquitin ligase, we asked whether praja2 regulates KSR2 ubiquitylation. Figure 2f shows that overexpression of wild-type praja2 markedly increased KSR2 ubiquitylation, whereas a praja2 inactive mutant (praja2rm) had no effect, thus indicating that a catalytically active protein was required for KSR2 ubiquitylation. The contribution of praja2 in KSR2 ubiquitylation was confirmed by the genetic silencing of praja2 in GBM cells. Indeed, downregulation of praja2 almost completely abrogated KSR2 ubiquitylation (Fig. 2g). Next, we sought to identify the lysine residue(s) of KSR2 that accept ubiquitin moieties by praja2. To this end, we took advantage of the available database reporting modified lysine residues on KSR2 identified by mass spectrometry analysis (Supplementary Fig. 2b). Based on this information,

we generated KSR2 mutants carrying the lysine acceptor site/s changed to arginine. We monitored ubiquitylation levels of either wild-type or lysine-mutants of KSR2 in serum-deprived or stimulated cells. The analysis identified lysine 281 as a relevant ubiquitin acceptor site on KSR2, since its mutation to arginine significantly reduced KSR2 ubiquitylation in response to serum stimulation (Supplementary Fig. 2b).

Ubiquitylated proteins often undergo proteolysis[38]. Accordingly, we evaluated whether praja2 regulates KSR2 stability. As shown in Fig. 2h, i, the expression of wild-type praja2, but not of its mutant praja2rm, decreased significantly the levels of KSR2 and reduced the half-life of co-expressed KSR2 (Supplementary Fig. 2c, d). Degradation of KSR2 induced by praja2 was mediated by the proteasome. Thus, pretreating cells with the proteasome inhibitor MG132 reversed the effects of praja2 on KSR2 stability (Fig. 2j, k). Furthermore, genetic silencing of praja2 in GBM cells significantly increased the steady-state levels of KSR2, supporting the role of praja2 in controlling KSR2 stability (Supplementary Fig. 3).

**Praja2 interrupts KSR2-AMPK oxidative pathway.** Since praja2 controls the stability of KSR2, we evaluated its role in the activation of its downstream effector AMPK. This kinase is activated by an elevated AMP/ATP ratio, which signals general stress caused by glucose deprivation or increased energetic demands. Phosphorylation of AMPKα at Thr172 present within the activation loop is required for kinase function[39]. Therefore, as a readout of kinase activity, we monitored phosphorylation of AMPKα at Thr172 using a specific anti-pThr172-AMPK antibody. As shown in Fig. 3a and b, glucose deprivation induced a time-dependent increase of AMPKα phosphorylation. Genetic silencing of praja2 markedly upregulated phosphorylation of AMPKα both under basal conditions or following glucose deprivation. We also monitored AMPKα

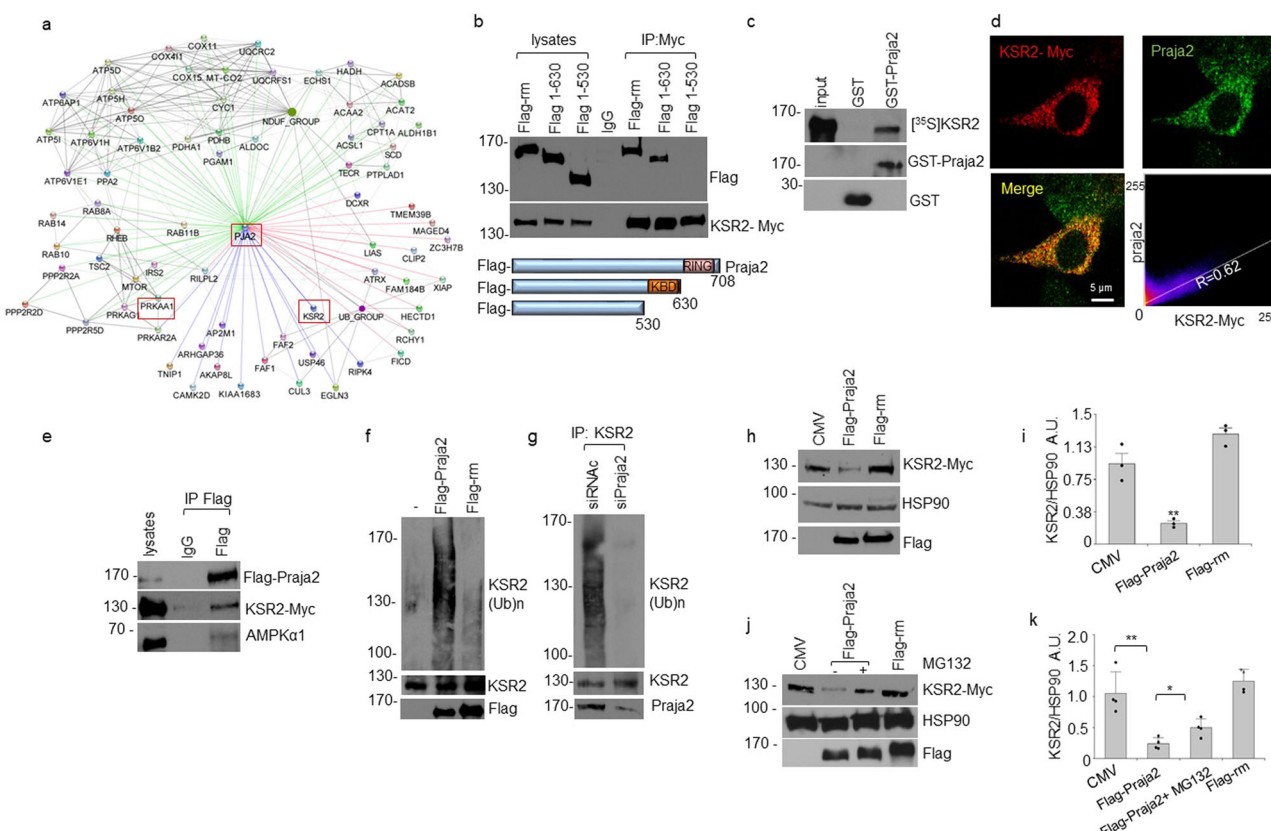

**Fig. 2 Assigning interaction of praja2 with other proteins possibly related to GBM. a** Praja2 protein–protein interaction (PPI) network. Praja2 complexes were purified from cell lysates of HEK293 cells transiently transfected with Flag-praja2rm vector or flag vector (control), which were independently subjected to an enrichment step with an anti-Flag derivatized resin, whose eluates were finally subjected to proteomic analysis. Identified praja2-binding partners involved in metabolic pathways and those reported in available databases were used to generate a PPI network. Connections are colored based on the interaction source: interactors identified in this study (green edges), CPTAC-GBM interactors (red edges), Intact interactors (blue edges), and STRING interactors (gray edges). The complete list of protein interactors is reported in Supplementary Data 1. **b** Schematic representation of praja2 constructs (lower panel) used in co-immunoprecipitation assays (upper panel). Lysates from HEK293 cells transiently transfected with Myc-tagged KSR2 and Flag-praja2 vectors were subjected to immunoprecipitation with an anti-Myc antibody. Precipitates and an aliquot of lysates were immunoblotted with anti-Flag and anti-Myc antibodies. **c** In vitro translated, [$^{35}$S] labeled KSR2 was subjected to pull-down assay with GST and GST-praja2 polypeptides. **d** U87MG cells were immunostained with anti-Myc and anti-praja2 antibodies and further analyzed by confocal microscopy. Scale bar = 5 μm. The Pearson's coefficient value of KSR2 and praja2 signals is shown (lower, right panel). **e** A trimeric complex composed of praja2, KSR2, and AMPKα1 was isolated from lysates of HEK293 cells transiently expressing Flag-praja2 and KSR2-Myc and subjected to immunoprecipitation with anti-Flag. **f** HEK293 cells transfected with HA–ubiquitin and Flag-praja2 or Flag-praja2rm and KSR2-Myc were treated for 6 h with 10 μM MG132. Lysates were immunoprecipitated with an anti-KSR2 antibody. Lysates and precipitates were immunoblotted with anti-HA, anti-Flag, and anti-KSR2 antibodies. **g** Lysates from growing HEK293 cells transiently transfected with HA–ubiquitin, KSR2-Myc, and control siRNA (siRNAc) or siRNAs targeting praja2 (siPraja2) were immunoprecipitated with anti-KSR2 antibody. Lysates and precipitates were immunoblotted with anti-HA, anti-praja2, and anti-KSR2 antibodies. **h** Cells were transiently transfected with flag-praja2 or flag-praja2rm for 24 h. Lysates were immunoblotted with the indicated antibodies. **i** Quantitative analysis of data shown in panel **h**. A mean value ± SEM of three independent experiments is reported. P value: * = 0.037; ** = 0.0026. **j** U87MG cells were transfected with a control vector or with vectors encoding for flag-praja2 or flag-praja2rm. Where indicated, cells were pretreated with MG132 for 6 h before harvesting. Lysates were subjected to immunoblot analysis with the indicated antibodies. **k** Quantitative analysis of data shown in panel **j**. A mean value ± SEM of four independent experiments is reported. P value: * = 0.011; ** = 0.0020.

phosphorylation using 5-aminoimidazole-4-carboxamide-1- β-D-ribonucleoside (AICAR), which is an AMP analog that activates AMPK independently of LKB1, a serine/threonine kinase that phosphorylates and activates AMPK[40]. In control GBM cells, AICAR efficiently increased AMPKα phosphorylation by several-fold over the control value (Fig. 3c, d). AICAR-induced phosphorylation of AMPKα was significantly upregulated in praja2-silenced cells. Downregulation of praja2 and KSR2 reversed, at least in part, the effects of praja2 silencing on AMPK (Fig. 3e, f).

Most cancers, including GBM, have a unique metabolic profile mostly based on aerobic glycolysis, also known as the Warburg effect[41,42]. The glycolytic profile of cancer cells is inhibited by AMPK that, once activated, suppresses glycolysis and inhibits

tumor growth[43]. Based on these observations, we monitored the metabolic profile of GBM cells by measuring the corresponding oxygen consumption rate (OCR) and extracellular acidification rate (ECAR). The assay was performed under basal conditions or in the presence of oligomycin (an ATP synthase inhibitor), carbonyl cyanide-4-(trifluoromethoxy) phenylhydrazone (FCCP) (a mitochondrial protonophore uncoupler), as well as of rotenone plus antimycin A (two mitochondrial transport chain inhibitors). Pharmacological treatment with inhibitors was used to discriminate basal and ATP-linked OCR. As expected, control GBM cells showed a predominant glycolytic profile, and the synthesis of ATP was linked to the glycolytic pathway (Fig. 3g–j and Supplementary Fig. 4). Genetic silencing of praja2 markedly

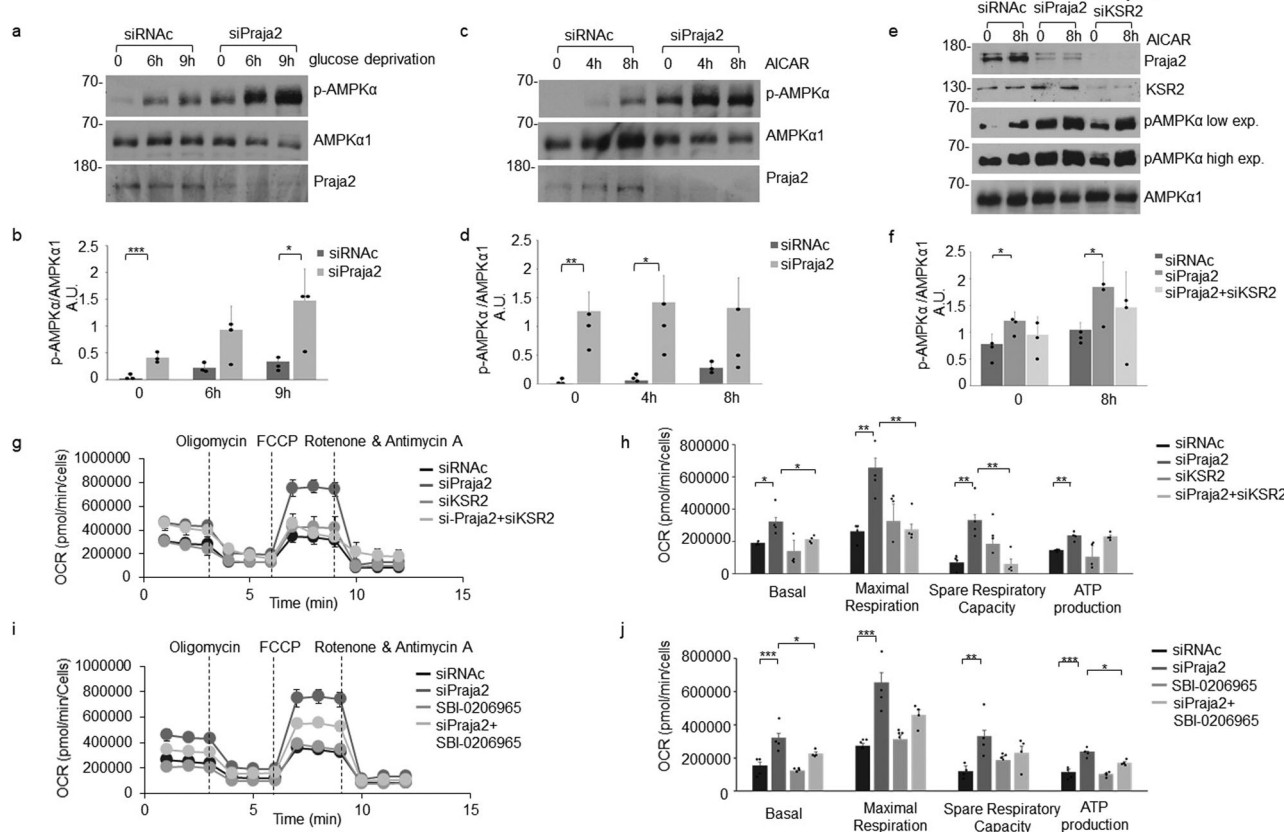

**Fig. 3 Praja2 restrains AMPK signaling and supports the glycolytic pathway. a** U87MG cells were transiently transfected with siRNAs (siRNAc or siPraja2). Twenty-four hours later, cells were left untreated (time point 0) or deprived of glucose for the indicated times. Lysates were immunoblotted with anti-pThr172-AMPKα, anti-AMPKα1, and anti-praja2 antibodies. **b** Quantitative analysis of the experiments shown in panel **a**. A mean value ± SEM of three independent experiments is reported. P value: *** = 0.00070; * = 0.028. **c** Same as in **a**, with the exception that glucose-supplemented cells were treated with 1 mM AICAR for the indicated times. **d** Quantitative analysis of the experiments shown in panel **c**. A mean value ± SEM of three independent experiments is reported. P value: ** = 0.0042; * = 0.014. **e** U87MG cells were transiently transfected with siRNAs (siRNAc, siPraja2 or siPraja2, and siKSR2). Twenty-four hours later, cells were left untreated (time point 0) or cells were treated with 1 mM AICAR for the indicated times. Lysates were immunoblotted with indicated antibodies. **f** Quantitative analysis of the experiments shown in panel **e**. A mean value ± SEM of three independent experiments is reported. P value: * = 0.014; * = 0.034. **g** Oxygen consumption rate (OCR) in U87MG cells transiently transfected with the indicated siRNAs. Reported data are the mean values ± SEM of four independent experiments. OCR was measured in real-time, under basal conditions, or in the presence of the indicated mitochondrial inhibitors: oligomycin, FCCP, antimycin A plus rotenone. **h** Indices of mitochondrial respiratory function, as calculated from the OCR profile of U87MG cells transiently transfected with the indicated siRNAs: basal OCR, maximal respiration, spare respiratory capacity, ATP production. Reported data were the mean values ± SEM of four measurements deriving from four independent experiments. P value: * = 0.011; * = 0.025; ** = 0.0069; ** = 0.0093; ** = 0.0061; ** = 0.0.0058 ; ** = 0.0026. **i** Oxygen consumption rate (OCR) in U87MG cells transiently transfected with reported siRNAs. When indicated, cells were pretreated with the AMPK inhibitor SBI-0206965 (5 μM for 4 h). Reported data were the mean values ± SEM of four independent experiments. **j** Indices of mitochondrial respiratory function, as calculated from the OCR profile of praja2-silenced and control U87MG cells: basal OCR, maximal respiration, spare respiratory capacity, ATP production. When indicated, cells were pretreated with the AMPK inhibitor SBI-0206965 (5 μM for 4 h). Reported data were the mean values ± SEM of four independent experiments. P value *** = 0.00014; * = 0.039; *** = 0.0.00013; ** = 0.0035; *** = 0.000090; * = 0.012.

enhanced the Spare Respiratory Capacity and the oxidative ATP production (total and basal rate), compared to control cells. Praja2 silencing had minor effects on the ECAR profile (Supplementary Fig. 4a, c, e, f), suggesting that the ligase mostly regulates oxidative phosphorylation. Since praja2 controls the KSR2 stability, we analyzed the contribution of KSR2 to the praja2-dependent oxidative pathway. Notably, concomitant downregulation of KSR2 reversed the effects of praja2 silencing on the respiratory capacity of GBM cells (Fig. 3g, h), without significant effects on ECAR (Supplementary Fig. 4e, f). Similarly, the inactivation of AMPK by a specific inhibitor (SBI-0206965) partially abrogated the effects of praja2 silencing on oxidative metabolism (Fig. 3i, j and Supplementary Fig. 4c, d). Minor effects on OCR were evident in GBM cells subjected to

downregulation of KSR2 or treatment with AMPK inhibitor alone (Fig. 3g–j). These findings indicate that praja2 sustains glycolysis in GBM and partly suppresses oxidative phosphorylation (OxPHOS). These effects are partly overlapping with those caused by KSR2 silencing. Moreover, the effect on glycolysis is also shown by the downregulation of pyruvate dehydrogenase (PDH) mRNAs (see below), a key enzyme of the glycolytic pathway.

**Praja2 induces a transcriptional metabolic rewiring of GBM cells.** To identify the gene networks and cellular pathways regulated by the *Praja2* gene, we investigated the effects of the praja2 knock-down in U87 glioblastoma cells by determining the RNA

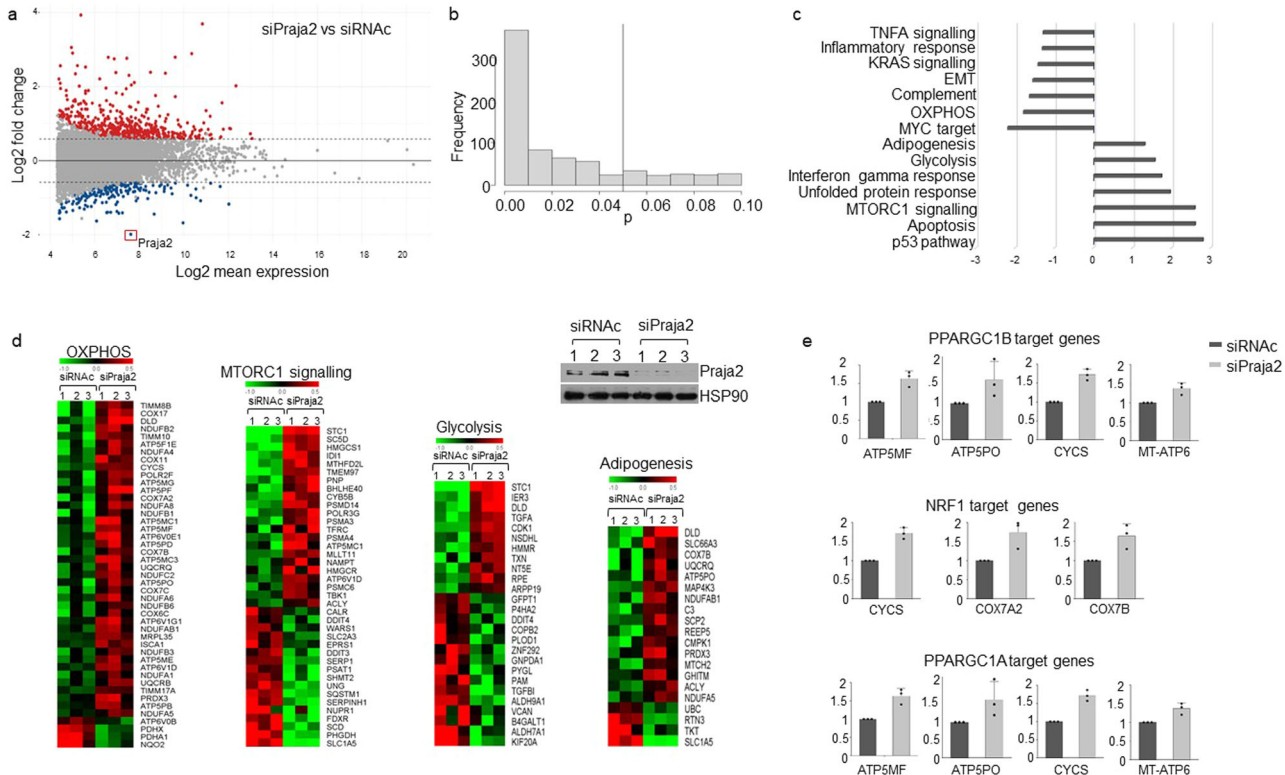

**Fig. 4 Transcriptional reprogramming of praja2-silenced GBM cells. a** MA-plot from RNA-Seq data analysis showing the transcriptome differences after praja2 silencing in U87 cells, compared to control. Red dots represent upregulated transcripts with fold-change ≥1.5 and adjusted $p$ value ≤0.10, while blue dots represent downregulated transcripts with fold-change ≤−1.5 and adjusted $p$ value ≤0.10. Gray dots represent those transcripts with −1.5< fold-change <1.5 and/or with an adjusted $p$ value >0.10. The x-axis represents the log2 of mean expression, while the y-axis represents the log2 fold-change, as computed by DESeq2. Dashed lines highlight the fold-change cutoff of ≥1.5 or ≤−1.5. **b** Histogram showing the distribution of adjusted $p$ values for the 725 differentially expressed genes. More than 81.5% of differentially expressed genes are associated with an adjusted $p$ value ≤ 0.05. **c** Histogram showing NES (normalized enrichment score) values of the molecular signatures statistically significant (FDR ≤0.25) involving the differentially expressed transcripts, as computed by the GSEA tool. **d** Heatmap summarizing expression data for the differentially expressed transcripts involved in the molecular signature of indicated pathways, as computed with GSEA, in siPraja2 versus siRNAc conditions. Normalized expression values in log2 scale and centered on the median value. Immunoblot analysis of praja2 in siRNA-transfected U87MG cells before RNA preparation is also shown. **e** IPA upstream regulator analysis of master upstream regulatory factors (PPARGC1A, PPARGC1B, and NRF1) on their target genes. Data were expressed as the ratio of the normalized read counts in siPraja2 vs siRNAc samples.

expression profiles by RNA sequencing (RNAseq). Reads were aligned to the reference human genome and identified about 10,200 transcripts in each biological replicate. Figure 4a, b show that 725 differentially expressed (DE) transcripts were found in praja2-silenced cells compared to negative controls. 526 RNAs were upregulated (fold-change ≥1.5 and adjusted $p$ value ≤0.10) and 199 downregulated (fold-change ≤−1.5 and adjusted $p$ value ≤0.10), including the praja2 RNA (fold-change = −3.961). To elucidate the molecular expression signature of praja2, we performed a Gene Set Enrichment Analysis (GSEA) for the differential expressed genes, using the Molecular Signature Database "Hallmarks" gene set collection[44]. Only those sequences showing a FDR ≤0.25 were selected for further analysis (Fig. 4c). We found a significant enrichment in functional categories involved in cell cycle regulation and apoptosis, cellular stress response, energetic metabolism, inflammation, and epithelial-to-mesenchymal transition (EMT) (Fig. 4c). In addition, this analysis revealed the presence of several differentially expressed transcripts encoding key components of signal transduction cascades frequently found deregulated in several human cancers, including mTOR, KRAS, and TNF-α/NF-kB signaling pathways (Fig. 4d). To analyze further the cascade of upstream transcriptional regulators, we performed an upstream regulator analysis (IPA). This method

exploits the known effects of master upstream regulatory factors on their target genes to pinpoint key actors responsible for the global gene expression changes detected in response to a given stimulus or condition, expressed as Z-score activation values. Based on Z-score activation values, we identified several transcriptional regulators, such as PPARGC1A, PPARGC1B, and NRF1, whose activity on target genes was enhanced by praja2 silencing (Fig. 4e and Supplementary Tables 1, 2).

**Development of transferrin-targeted nanoparticles for delivery of siRNA targeting praja2.** The data above indicate that praja2 is upregulated in high-grade glioma and acts as an important regulator of the transcriptional program underlying GBM growth and metabolism. This led us to investigate if downregulation of praja2 in growing GBM lesions reduces tumor expansion. To this end, we took advantage of a recently developed strategy to deliver drugs/molecules in vivo based on the use of self-assembling nanoparticles (SANPs)[45]. SANPs deliver anionic-charged drugs and nucleic acids, including RNA molecules, to different tissues and tumors in vivo, in some cases overcoming drug resistance of highly aggressive human cancers[46–48]. SANPs formulations are stable in BSA and serum and possess a low hemolytic activity and

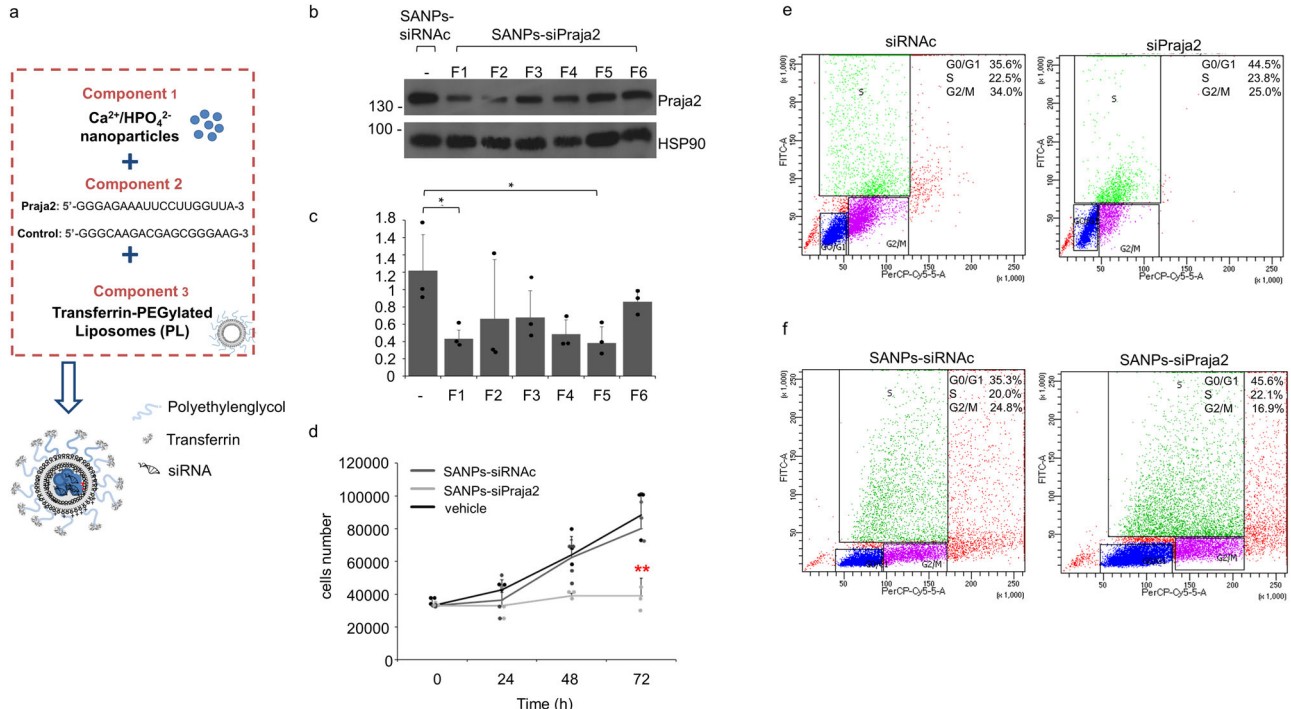

**Fig. 5 Generation of SANPs-siRNA targeting praja2. a** Schematic representation of self-assembling nanoparticles (SANPs) conjugated with transferrin and encapsulating control siRNA or siRNA targeting praja2. **b** Cells were perfused for 72 h with different preparations of SANPs-siRNAs targeting praja2 (F1: SANP1-siPraja2; F2: SANP2-siPraja2; F3: SANP3-siPraja2; F4: SANP4-siPraja2; F5: SANP5-siPraja2; F6: SANP6-siPraja2) or control SANPs (SANPs-siRNAc). Lysates were immunoblotted with anti-praja2 and anti-HSP90 antibodies. **c** Quantitative analysis of the experiments shown in panel **b**. Reported data were the mean values ± SEM of three independent experiments. P value: * = 0.0167; * = 0.0169. **d** Growth curves of U87MG cells treated with vehicle, SANPs-siRNAc, SANPs-siPraja2. At indicated time points, cells were harvested and counted. Three independent experiments were performed and the corresponding mean values ± SEM are shown. P value: ** = 0.0054. **e**, **f** FACS analysis of U87MG cells treated with siRNAc or siPraja2 (**e**), and SANPs-siRNAc or SANPs-siPraja2 (**f**) for 72 h. Cell cycle distribution (G0/G1, S, and G2/M) of treated cells is indicated as a percentage of total cells scored.

reliable biodistribution values in tissues and the whole brain. SANPs showed a diameter ranging from about 115 to 160 nm and all formulations were characterized by a narrow size distribution (PI <0.2), and a positive surface charge (ζ from about 14.4 to 36.8 mV) (Supplementary Table 4). SANPs with different lipid compositions and containing siRNA targeting human praja2 were prepared (Supplementary Table 3). The encapsulation of siRNAs in all SANPs formulations led to an increase in the size that remained <160 nm in the majority of the SANPs (Supplementary Tables 4, 5). The use of different lipid molar ratios did not significantly affect the particle size. Only in the case of SANPs3-siPraja2, containing the cationic lipid DC-CHOL/DSPE-PEG, the size of the vesicles was higher (about 200 nm). PI of the nano-vectors was <0.2 in all cases, except for the formulation based on DC-CHOL/DSPE-PEG (SANP3-siPraja2) characterized by a more heterogeneous lipid particle population (PI >0.3). Furthermore, the encapsulation of siRNA led in all cases to a reduction of the zeta potential value, due to the presence of molecules of negatively charged siRNAs on the SANPs surface. Finally, the encapsulation efficiency of both siRNAs, (siPraja2 and siRNAc) resulted between 90 and 100%.

The biological activity of SANPs-siPraja2 was assessed in vitro by monitoring praja2 levels in cultured GBM cells exposed to different preparations of SANPs-siPraja2. Figure 5b, c show that treatment with SANPs-siPraja2 (preparations SANP1-SANP6) for 72 h was the most efficient in downregulating praja2 levels, compared to SANPs carrying control siRNA. In the experiments in GBM cells and in vivo, we decided to use the nanoparticle preparation SANP1-Praja2. To reduce the amount of SANPs

needed for the in vivo experiments, we optimized the encapsulation of siPraja2 into SANP1 with a higher siRNA loading (185 μg/mg lipids). SANP conjugation with siRNA did not significantly affect the characteristics of highly-loaded SANPs in terms of actual loading and mean diameter (Supplementary Table 5). Transferrin ligand was then conjugated on the surface of SANPs 1-Praja2 to support the optimal transfer of siRNA-containing SANPs across the blood–brain barrier (BBB) via receptor-mediated transcytosis (Fig. 5a). The conjugation of SANPs with transferrin (SANPs 1-(2)Tf Praja2) did not significantly influence the physico-chemical characteristics of SANPs, with the exception of a slight increase of the PI value (0.3) reasonably attributed to the presence of transferrin at the lipid nanoparticles surface. First, we probed the role of praja2 in cell growth by determining the proliferation rate of GBM cells transiently transfected with siRNAs targeting praja2 or treated with SANPs 1-(2)Tf Praja2. Figure 5d and Supplementary Fig. 5 show that downregulation of praja2 markedly affected the growth rate of GBM cells. The analysis was complemented by monitoring the cell cycle progression of GBM cells by fluorescent-activated cell sorter (FACS) analysis. The results shown in Fig. 5e, f confirmed that praja2 silencing, either by siRNAs transfection or treatment with SANPs 1-(2)Tf Praja2, significantly inhibited the proliferation rate of GBM cells, inducing a growth arrest at the G0/G1 phase of the cell cycle.

**Brain delivery of SANPs-siPraja2 suppresses GBM growth**. The data reported above indicate that praja2 as E3 ubiquitin ligase

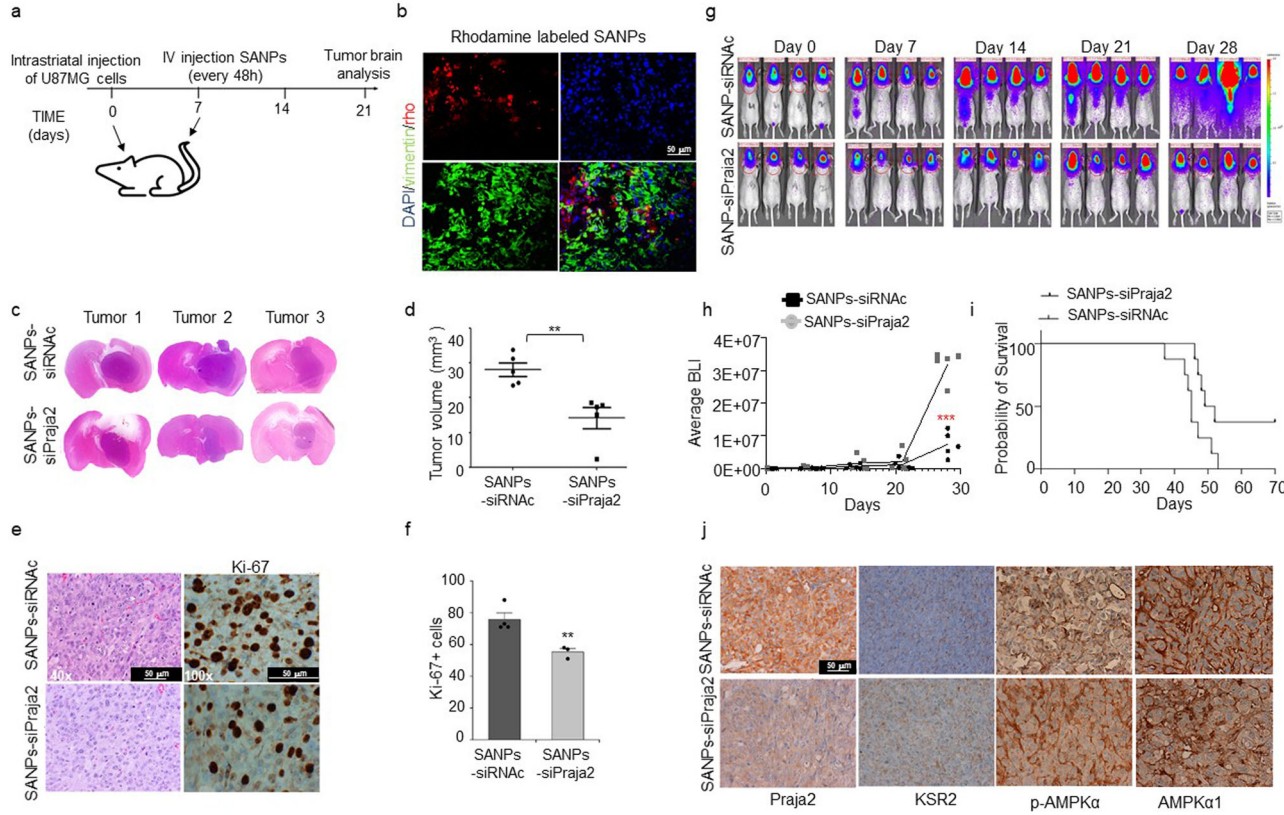

**Fig. 6 Systemic delivery of SANPs-siPraja2 inhibits GBM growth. a** Schematic view of the experimental procedures. U87MG cells were stereotaxically implanted into the brain of nude mice (time 0). One week post-implantation, SANPs-siRNAs were i.v. injected into the caudal vein every 48 h, for a total of 14 days of treatment. At 3 weeks post-implantation, mice were sacrificed and tumor lesions isolated and further characterized. **b** Brain distribution of rhodamine-labeled SANPs-siPraja2 by fluorescence analysis at 9 h after i.v. injection. GBM lesions were identified by immunostaining the same brain sections with an anti-human vimentin antibody. Nuclei were stained with DAPI. Representative images are shown. Scale bar, 50 μm. **c** Tissue sections from tumor lesions were stained with hematoxylin/eosin. **d** Quantitative analysis of the tumor volume is expressed as a mean value ± SEM. Three independent experiments were performed. *P* value: ** = 0.0014. **e** Tumor sections were stained with hematoxylin/eosin or immunostained for Ki-67. Scale bar, 50 μm. **f** Quantitative analysis of Ki-67-positive cells in tumor lesions from control and SANPs-siPraja2 treated mice. The data were expressed as a mean value ± SEM from three independent experiments. *P* value: ** = 0.0055. **g** U87MG-Luc cells were injected into the brain of 6 weeks old CD1 mice. Three hours following implantation, bioluminescent intensity (BLI) was measured by intraperitoneal injection of 150 mg/kg ᴅ-Luciferin potassium salt. At 1-week post-injection, based on BLI measurement, mice were randomized into two experimental groups of 12 animals, and each group was treated by tail vein injection with SANPs-siPraja2 (GP1) and SANPs-siRNAc (GP2), respectively. Treatments were repeated twice a week for 4 weeks, then four mice for each group were sacrificed and organs collected. BLI analysis was performed every week and quantitative data were collected. A representative set of animals for each experimental group is shown. **h** Quantitative and cumulative analysis of BLI scores. *** <0.001. **i** Kaplan–Meier curve of treated animals. At 52 days from U87MG implantation, all the animals from SANPs-siRNAc group died. In contrast, about 40% of SANPs-siPraja2 mice were still healthy, but the experiment was terminated in accordance with Authorities guidelines. **j** Immunostaining analysis for praja2, KSR2, pThr172-AMPKα, and AMPKα1 in tumor sections from control and SANPs-siPraja2 treated mice. Scale bar, 50 μm.

controls critical regulators of metabolic pathways and also the transcriptional reprogramming of transformed cells, thus becoming a potential molecular target for GBM therapy. We tested this hypothesis by analyzing the inhibition in vivo of GBM growth by SANPs 1-(2)Tf Praja2 nanoparticles. Firstly, we evaluated the biodistribution and accumulation of the nanoparticles within a milieu of a growing GBM. To this aim, we used orthotopic xenograft models of human GBM, where U87MG cells are stereotaxically implanted in the subventricular brain area of immune-compromised mice[29] (Fig. 6a). After the first week of tumor engraftment, transferrin-targeted and rhodamine-labeled SANPs-siPraja2 were perfused in mice tail vein. Animals were sacrificed at 9 h post-injection, and isolated fresh brain sections were analyzed by fluorescence microscopy using the OCT instrument. As shown in Fig. 6b, nanoparticles accumulated within the tumor lesions at 9 h post-perfusion, indicating that SANPs represent a suitable 'carrier' of molecules for the passage through the BBB.

Next, we tested if the treatment with SANPs 1-(2)Tf Praja2 (SANPs-siPraja2 in the following part of the text and figures) inhibits GBM in vivo. U87MG cells were stereotaxically implanted in the left caudate nucleus of the mouse brain. At 1-week post-implantation, mice bearing orthotopic GBM lesions were perfused every two days with either SANPs-siPraja2 or SANPs-siRNAc. Three weeks after the first SANPs perfusion, mice were sacrificed, and brain lesions were analyzed. As shown in Fig. 6c, d, treatment with SANPs-siPraja2 significantly reduced the tumor volume, compared to controls (SANPs-siRNAc). Hematoxylin/eosin staining showed a GBM cell population with a pleomorphic volume and morphology; the nuclei appeared voluminous, vesicular, and compacted, showing condensed nuclear chromatin with a prominent nucleolus (Fig. 6e). Cells were arranged in mutual contact in a disordered fashion, but they were devoid of cohesion. The more relevant histologic difference between tumor lesions of both experimental groups was a significant reduction of cells in tumor sections of SANPs-

siPraja2-treated mice, compared to controls. A notable reduction of the proliferation marker Ki67 in tumor sections from SANPs-siPraja2-treated mice was also evident (Fig. 6e, f).

To dynamically assess the inhibitory effects of RNAi-nanoparticles on GBM growth, we repeated the experiments using a U87MG cell line stably expressing the luciferase gene (U87MG-Luc). This is a widely used enzymatic bioluminescence system that allows the tracking of Luc-expressing cells in vivo in a quantitatively dynamic manner[49]. U87MG-Luc cells were stereotaxically implanted in the subventricular brain area of nude mice. At 1-week post-implantation, mice were treated with SANPs-siPraja2 or SANPs-siRNAc three times/week for a total time of 4 weeks of treatment. Bioluminescence intensity (BLI) data were collected overtime period and analyzed. As reported in Fig. 6g, h and Supplementary Fig. 6a, RNAi-SANPs treatment for 4 weeks dramatically inhibited tumor growth, as shown by a marked decrease of bioluminescence intensity (BLI) in SANPs-siPraja2-treated mice, compared to controls. A Kaplan–Meier analysis of both experimental groups showed an increased survival rate of SANPs-siPraja2-treated mice, compared to controls (Fig. 6i). Biochemical tests on plasma samples from nanoparticles-treated mice showed no major effects of the treatment on kidney and liver functions (Supplementary Fig. 6b). Immunohistochemical analysis confirmed the downregulation of praja2 by SANPs-siPraja2 treatment (Fig. 6j). As predicted, the staining for KSR2, pThr172-AMPK, and AMPKα1, was increased in tumor sections from SANPs-siPraja2-treated mice, compared to controls (Fig. 6j).

## Discussion
In this study, we report the identification of a regulatory gene network controlled by praja2 that underlies cancer cell metabolic reprogramming and glioblastoma growth. Expression of praja2 selectively marks wild-type IDH1-positive human glioma. Protein–protein interaction analysis identified the kinase scaffold KSR2 as a novel partner of praja2-assembled macromolecular complexes, and a direct target of the ubiquitin-proteasome system. By regulating KSR2 ubiquitylation and stability, praja2 controls downstream activation of AMPK and, as a consequence, oxidative phosphorylation and cancer cell metabolism. Nanoparticle-mediated silencing of praja2 in vivo markedly inhibited GBM growth and significantly improved the survival rate of treated mice.

Glioblastoma is one of the most lethal and untreatable human cancer worldwide with a poor rate of patient survival, highlighting the urgent development of novel therapeutic approaches for the disease[1]. GBM cells are characterized by a profound genetic reprogramming that supports the metabolic switch from oxidative respiration to the glycolytic pathway (Warburg effect). The advantage of this metabolic change is mostly based on the consumption of large amounts of glucose to produce intermediates needed for the synthesis of building blocks in rapidly-dividing cancer cells. The mechanisms underlying this metabolic reprogramming have been largely explored and functionally characterized[42,50,51]. In this context, AMPK is the principal metabolic sensor that also contributes to the maintenance of cellular homeostasis, thus promoting catabolic processes while inhibiting the anabolic pathways. Beyond energy homeostasis, AMPK regulates cellular responses to environmental stresses (hypoxia and chemicals), ROS/redox balance, autophagy, apoptosis, cell proliferation, cellular architectures, and mitochondrial activities[52,53]. Metabolically, once activated, AMPK promotes mitochondrial biogenesis and oxidative ATP synthesis through the transcriptional activation of genes encoding for components of the respiratory chain machinery[53]. Furthermore, stimulation of

AMPK-mediated metabolic pathways, including oxidative glucose consumption, thermogenesis, and fatty acid oxidation requires KSR2[33,37]. Knockout mice carrying deletions of KSR2 or AMPKα2 subunit showed a similar metabolic phenotype, which is characterized by a drastic reduction of energy expenditures, impaired oxidative pathways, higher lipogenesis and glucose intolerance[54]. This metabolic phenotype replicates the clinical features of obese young patients carrying KSR2 mutations characterized by a reduced heart rate, downregulation of basal metabolism, and severe insulin resistance[37]. These findings support the model whereby the KSR2-AMPK axis works at the intersection between cell proliferation and metabolic pathways, adjusting the energy production to increase the metabolic needs of proliferating cells. However, the crosstalk and interplay between the ubiquitin system and the KSR2-AMPK pathway were largely unknown and, here, we provide a link between metabolism and growth signaling. We identified KSR2 as a novel partner and substrate of a praja2-UPS pathway. By controlling KSR2 stability, praja2 regulates the glycolytic metabolism that supports cancer cell proliferation. Thus, downregulation of praja2 in GBM cells abolished KSR2 ubiquitylation, promoting the switch from aerobic glycolysis to oxidative metabolism. The effects of praja2 were reversed by concomitant KSR2 silencing, indicating that the KSR2-AMPK axis is, indeed, a relevant functional target of praja2. This was supported by the finding that AMPK activation under glucose deprivation or agonist stimulation was markedly induced by praja2 downregulation. However, AMPK inhibition only partially reversed the metabolic effects of praja2 silencing, suggesting that the ligase-KSR2 pathway can also operate through AMPK-independent mechanisms. This hypothesis was supported by proteomic data reported in this study, which show the presence of a variety of praja2 protein-binding partners and putative substrates involved in different aspects of oxidative and glycolytic pathways. We propose that praja2 is a novel upstream regulator of the metabolic axis controlled by the KSR2-AMPK signaling unit, functionally coupling and adapting growth to the metabolic needs of rapidly-dividing cells. In GBM cells, upregulation of praja2 contributes to reduce the AMPK-dependent oxidative metabolism and switching on glycolysis. Inactivation of the LKB1-AMPK oncosuppressive pathway observed in GBM lesions further supports this conclusion[29,55]. This was also shown by RNA analysis. Thus, mRNA transcripts encoding for proteins of the oxidative respiratory chain were markedly upregulated in praja2-silenced GBM cells. In the same cells, upstream regulatory factors of mitochondrial transcription and downstream effectors of the LKB1-AMPK pathway, such as PPARG1A, PPARG1B, and NRF1, were induced. Moreover, RNA analysis in praja2-silenced cells identified also a variety of gene products that cooperate and promote cancer growth, highlighting a general role of praja2 in cell growth. This aspect has been previously clarified by the demonstration that praja2 inhibits the oncosuppressive Hippo pathway[29,56]. This antiproliferative effect is also reinforced by the praja2-mediated downregulation of KSR1, the homolog of KSR2, and the principal effector of the Ras-Raf-MEK mitogenic pathway[21,22]. Preventing proteolysis of KSR1 by praja2 impairs growth and promotes embryonic stem cell differentiation[21]. Our data indicate that praja2 by modulating the KSR2-AMPKα signaling pathway controls the oxidative metabolism of GBM cells. However, the mechanism(s) underlying the transcriptional regulation of PPARG1A, PPARG1B, and NRF1 genes involved in the mitochondrial metabolic rewiring by praja2 needs further investigation.

The essential role of praja2 in cancer cell growth suggests that this ligase constitutes a valid therapeutic target for GBM treatment. We tested this hypothesis in vivo by monitoring GBM growth in mice perfused with transferrin-coated lipid

nanoparticles delivering RNAi molecules targeting praja2 to the brain. This treatment substantially inhibited GBM growth and significantly improved the survival rate of treated mice. Perfusing lipid particles in peripheral blood often results in unhealthy reactions due to the toxic effects of treatment[57]. To avoid these unnecessary complications, we modified the protocol and generated small-sized SANPs with increased stability in the serum, low hemolytic activity, and reliable biodistribution in tissues and the whole brain. Peripheral administration of SANPs to the animals had no significant off-target effects on vital organs, such as bone marrow, kidney and liver, supporting a potential therapeutic use of our approach for future studies. Moreover, SANPs have been designed for an easy scale-up process in order to speed up the technology transfer from the bench to the bed side[45]. Here, we proposed a strategy based on transferrin-targeted self-assembling nanoparticles (SANPs) to deliver inhibitory RNA molecules to the brain. The use of this strategy can be also considered for the treatment of other disorders, such as chronic kidney disease, where praja2 has been shown to play a significant pathogenic role in this clinical condition[31].

An important aspect that needs to be further addressed is whether downregulation of praja2 by the RNAi-nanoparticles potentiates the efficacy of other therapeutic strategies, such as chemotherapy and/or radiotherapy that are currently being used for GBM treatment. The contribution of praja2 in the early steps of glial cell transformation and its role in the induction and maintenance of cancer stemness, an acquired feature of aggressiveness and invasive potential of GBM cells, are important issues that need to be further explored. Notably, praja2 levels were selectively increased in primary GBM lesions carrying wild-type IDH1 gene. This finding contributes to further define the molecular characterization and the differential regulation of transcriptional programs and metabolic pathways operating in GBM lesions. However, the mechanisms controlling praja2 expression in different glioma lesions and its impact on cancer cell reprogramming are aspects that need further investigations.

In conclusion, we identified praja2 as a novel marker of wild-type IDH1-positive glioma and an important central regulatory element of the metabolic gene network controlled by the KSR2-AMPK axis in GBM cells. By regulating KSR2 stability and downstream AMPK signaling, praja2 supports cancer cell metabolic reprogramming and tumor growth. Targeting praja2 in vivo, by reversing the mitochondrial metabolic rewiring of growing cancer cells, appears a viable opportunity to optimize the therapy of GBM.

## Methods

**Cell culture**. Human glioblastoma cells (U87MG and U87MG-Luc) and human embryonic kidney cells (HEK293) were obtained from the American Type Culture Collection (Manassas, Va.). Cells were cultured in Dulbecco's Modified Eagle Medium supplemented with 10% fetal bovine serum (Gibco™ Fetal Bovine Serum South America, Thermo Scientific Fisher-US), 2 mM L-glutamine, 50 U/ml penicillin, 50 µg/ml streptomycin, at 37 °C, 5% $CO_2$ and 95% of humidity.

**Plasmid and transfection**. Vectors encoding praja2 and HA–ubiquitin were previously described[15,19]. In particular, praja2 inactive RING mutant (praja2rm) carries cysteines 634/637 changed to alanine[19]. KSR2-Myc was purchased from Genscript. siRNAs targeting praja2 and KSR2 were transfected using Lipofectamine 2000 (Invitrogen). For praja2 silencing experiments, we used a pool of four siRNAs and sequences (Dharmacon) are the following: sequence 1: 5′-GAGAUGA-GUUUGAAGAGUU-3′; sequence 2: 5′-GGGAGAAAUUCCUUGGUUA-3′; sequence 3: 5′-UGACAAAGAUGAAGAUAGU-3′; sequence 4: 5′-UCAGAU-GACCUCUUAAUAA-3′. KSR2 siRNA sequence (Thermo Fisher) is 5′-AAAUG-CUGAAGAGUCCAAAGUCCGU-3′. Control siRNA was purchased from Ambion (am4637). For the cell growth curve, two independent siRNA mixtures were used: mix 1 (siRNA1 + siRNA3) and mix 2 (siRNA2 + siRNA4).

**Immunoprecipitation and immunoblot**. Cells were lysed with 1% Triton buffer (150 mM NaCl, 50 mM Tris-HCl pH 7.5, EDTA 1 mM, 1% Triton, 5 mM

$MgCl_2$) supplemented with protease inhibitors, phenylmethylsulfonyl fluoride (PMSF) and phosphatase inhibitors. Lysates were incubated overnight with the indicated antibodies for immunoprecipitation; then, pellets were washed three times with lysis buffer. Precipitates and a quote of lysates were loaded on SDS polyacrylamide gel and blotted on nitrocellulose membrane. Filter were blocked with 5% milk in TBS-Tween 0.1% and incubated with primary antibodies overnight. After incubation with secondary antibodies, proteins were detected with ECL.

**In vitro pull-down assay**. GST-fusions were expressed and purified from BL21 (DE3) pLysS cells. For in vitro-binding assays, 20 µl of GST and GST-Praja2 beads were incubated in in vitro-translated, [$^{35}$S]-labeled KSR2 in 200 µl lysis buffer (150 mM NaCl, 50 mM Tris-HCl pH 7.5, 1 mM EDTA, 0.5% Triton X-100) in rotation at 4 °C overnight. Pellets were washed four times in lysis buffer supplemented with NaCl (0.4 M final concentration) and eluted in Laemmli buffer. Eluted samples were size-fractionated on SDS-PAGE and immunoblotted or subjected to autoradiography.

**Immunofluorescence and confocal analysis**. Cells plated on cover glass were fixed with paraformaldehyde 5% (Merck), permeabilized with 0,3% Triton (Merck), blocked with BSA 5% (SERVA), and finally stained with the indicated primary antibodies. Signals were revealed with rhodamine-conjugated secondary antibodies (1:200, Invitrogen) and nuclei were stained with DAPI. Immunostaining was visualized using Zeiss LSM700 confocal microscope.

**Materials and antibodies**. The following chemicals were used: 1,2-dioleoyl-3-trimethylammonium-propane chloride (DOTAP), N-palmitoyl-sphingosine-1 {succinyl[methoxy (polyethylene glycol)$_{2000}$]} (cer-PEG), 3ß-[N-(N′,N′-dimethyla-minoethane)-carbamoyl]cholesterol hydrochloride (DC-chol), cholesterol (CHOL), 1,2-distearoyl-sn-glycero-3-phosphoethanolamine-N-[amino(polyethylene glycol)-$_{2000}$] (DSPE-PEG$_{2000}$), 1,2-distearoyl-sn-glycero-3-phosphoethanolamine-N-[amino(polyethylene glycol)-2000]-maleimide (DSPE-PEG-Mal), human transferrin (Tf), ammonium ferrithiocyanate, sodium borate, sodium chloride, sodium citrate, sodium phosphate, HEPES, ethylenediaminetetraacetic acid (EDTA), citric acid, Sepharose G-25, and 2-iminothiolane (Traut's reagent) 1,2-dioleoyl-sn-gly-cero-3-phosphoethanolamine-N-(lissamine rhodamine B sulfonyl ammonium salt) (Rhod), were purchased from Spectra2000 s.r.l. (Rome, Italy). Sodium chloride, sodium phosphate dibasic, calcium chloride, potassium chloride, were obtained from Merck Life Science s.r.l (Milan, Italy). Forskolin (#F3917) was purchased from Merck; AICAR (#A611700) was obtained from Toronto research chemicals; AMPK inhibitor SBI-0206965 was purchased from SIGMA (SML1540). The following primary antibodies were used: Flag (1:2000 immunoblot, 1:200 immuno-precipitation; #F3165, Merck); HA.11 (1:1000; #16B12, Biolegend); praja2 (1:1000 immunoblot, 1:200 immunofluorescence; #A302-991A, Bethyl Laboratories); phospho-Thr172-AMPKα (1:1000; #2535 S, Cell Signaling); AMPKα (1:1000; ♯07350, Merck); Myc (1:1000 immunoblot, 1:500 immunofluorescence; #M4439, Merck); KSR2 (1:50 immunoprecipitation; #ab173377, Abcam); KSR2 (1:1000 immunoblot, 1:50 immunoprecipitation; #sc100421, Santa Cruz); HSP90 (1:5000 #60318-1-Ig, Proteintech); MOB1 (1:50, #ab236969, Abcam). Antibody-antigen complexes were detected by HRP-conjugated antibodies (Biorad) and ECL (Euroclone).

**Proteomic analysis of praja2 protein complexes**. HEK293 cells overexpressing Flag-praja2rm were harvested and lysed with a buffer containing 1% Triton X-100, 150 mM NaCl, 1 mM EDTA, and 50 mM Tris-HCl (pH 7.5) and supplemented with protease inhibitors, PMSF and phosphatase inhibitors. Lysates were immuno-precipitated with anti-Flag M2 affinity gel (Merck, # A2220) for 3 h. After three washes using lysis buffer, proteins were eluted by incubation with 3xFlag-peptide (Thermo Fisher, #A36805) 150 ng/µl in PBS, for 2 h. As a control, immunoprecipitation of lysates from HEK293 cells overexpressing Flag-empty vector was performed. Immunopurified proteins were analyzed by 10% T SDS-PAGE. After staining with colloidal Coomassie blue, whole gel lanes were cut in 15 slices, minced, and washed with water. Corresponding proteins were separately in-gel reduced, S-alkylated with iodoacetamide, and digested with trypsin, as previously reported[58]. Individual protein digests were then analyzed with a nanoLC-ESI-Q-Orbitrap-MS/MS platform consisting of an UltiMate 3000 HPLC RSLC nano system (Thermo Fisher Scientific, USA) coupled to a Q-ExactivePlus mass spectrometer through a Nanoflex ion source (Thermo Fisher Scientific). Peptides were loaded on an Acclaim PepMap™ RSLC C18 column (150 mm × 75 µm ID, 2 µm particles, 100 Å pore size) (Thermo Fisher Scientific), and eluted with a gradient of solvent B (19.92/80/0.08 v/v/v water/acetonitrile/formic acid) in solvent A (99.9/0.1 v/v water/formic acid), at a flow rate of 300 nl/min. The gradient of solvent B started at 3%, increased to 40% over 40 min, raised to 80% over 5 min, remained at 80% for 4 min, and finally returned to 3% in 1 min, with a column equilibrating step of 30 min before the subsequent chromatographic run. The mass spectrometer operated in data-dependent mode using a full scan (m/z range 375–1500, a nominal resolution of 70,000, an automatic gain control target of 3,000,000, and a maximum ion target of 50 ms), followed by MS/MS scans of the ten most abundant ions. MS/MS spectra were acquired in a scan m/z range 200–2000, using a

normalized collision energy of 32%, an automatic gain control target of 100,000, a maximum ion target of 100 ms, and a resolution of 17,500. A dynamic exclusion value of 30 s was also used. Triplicate analysis of each sample was performed to increase the number of identified peptides/protein coverage.

MS and MS/MS raw data files per lane were merged for protein identification into Proteome Discoverer v. 2.4 software (Thermo Scientific), enabling the database search by Mascot algorithm v. 2.6.1 (Matrix Science, UK) with the following parameters: UniProtKB human protein database (11/2020, 214889 sequences) including the most common protein contaminants; carbamidomethylation of Cys as fixed modification; oxidation of Met, deamidation of Asn and Gln, and pyroglutamate formation of Gln as variable modifications. Peptide mass tolerance and fragment mass tolerance were set to ±10 ppm and ±0.05 Da, respectively. Proteolytic enzyme and a maximum number of missed cleavages were set to trypsin and 2, respectively. Protein candidates assigned on the basis of at least two sequenced peptides and a Mascot score ≥30 were considered confidently identified. Definitive peptide assignment was always associated with manual spectra visualization and verification. Results were filtered to a 1% false discovery rate. A comparison with results from the corresponding control allowed to identify contaminant proteins in each experiment that, nonetheless their abundance, were removed from the list of praja2-interacting partners (Supplementary Data 1).

**Protein–protein interaction (PPI) network analysis**. The protein network representing praja2 interactors was built starting from three sets of interaction data. The first set represents praja2 interactors directly identified in this study by proteomics. The 1538 interactors identified were filtered for proteins involved in metabolic pathways by using the gene IDs reported in KEGG pathways (KEGG:hsa04152, KEGG:hsa00190, KEGG:has_M00087, KEGG:hsa01212, KEGG:hsa00010), thus obtaining a network of 55 nodes and 54 edges. The second set of interactions was built by using all the 24 praja2 direct interactors reported in the Intact database [ftp://ftp.ebi.ac.uk/pub/databases/intact//2021-07-06/psimitab/intact.txt]. The third set of interactions was built starting from CPTAC-GBM discovery cohort protein assay data, composed of 99 samples and 10,409 proteins. Data used in this work were generated by the Clinical Proteomic Tumor Analysis Consortium (NCI/NIH)[59]. Starting from the protein abundance matrix, the ARACNE algorithm was applied from the MINET package with default parameters and an interaction network composed of 10,409 nodes and 61,116 edges was generated[60]. Then, the subnetwork of 13 ARACNE interactors of praja2 was extracted. The three networks were loaded into Cytoscape version 3.7.2 and the STRINGify function from the stringApp tool was applied to each network[61,62]. Then, networks were merged using the Cytoscape merge function and performing graph union.

**Total RNA extraction, libraries preparation, and sequencing with Illumina technologies**. U87MG glioblastoma cells have been transfected, in biological triplicate, with a siRNA targeting the praja2 gene or control siRNA. Libraries preparation was performed as described previously[63]. Cells were collected and homogenized in Trizol Reagent. Total RNA extraction was performed with an RNA Clean and Concentrator Kit (Zymo Research Corp., Irvine, CA, USA), following the manufacturer's instructions. RNA purity and integrity were assessed with a Nanodrop 2000c spectrophotometer (Thermo Fisher Scientific, Waltham, MA, USA) and a 4200 TapeStation instrument (Agilent Technologies, Santa Clara, CA, USA), respectively. For RNA purity, an A260/280 ratio of ~2.0 and an A260/230 ratio of 2.0–2.2 were considered acceptable; for RNA integrity, an RNA Integrity Number (RIN) of 9.0–10.0 has been obtained for all samples, indicating the absence of degradation and high integrity of RNA samples. For a precise estimation of the RNA concentration, a Qubit 2.0 fluorometer assay (Thermo Fisher Scientific, Waltham, MA, USA) was used. For RNA sequencing, 1 μg of high-quality total RNA was used for library preparation with a TruSeq Stranded Total RNA Sample Prep Kit (Illumina, San Diego, CA, USA) and sequenced (paired-end, 2 × 75 cycles) on the NextSeq 500 platform (Illumina, San Diego, CA, USA).

**RNAseq data and Gene Set Enrichment Analysis (GSEA) analysis**. Data analysis was performed as described previously[64]. In detail, the raw sequence files generated (.fastq files) underwent quality control analysis using FASTQC (http://www.bioinformatics.babraham.ac.uk/projects/fastqc/) and adapter sequences were removed using Trimmomatic version 0.38[65]. Filtered reads were aligned on the human genome (assembly hg38) considering genes present in GenCode Release 36 (GRCh38.p12) using STAR v2.7.5a with standard parameters[66]. Quantification of expressed genes was performed using featureCounts[67] and differentially expressed genes were identified using DESeq2[68]. A given RNA was considered expressed when detected by at least ≥10 raw reads. Differential expression was reported as | fold-change|(FC) ≥1.5 along with associated adjusted $p$ value ≤0.10, which was computed according to Benjamini–Hochberg. About 10,200 transcripts were expressed in each biological replicate of the two conditions, setting a normalized read-count cutoff ≥10, while DESeq2 highlighted 725 transcripts differentially expressed with 526 transcripts upregulated (fold-change ≥1.5 and adjusted $p$ value ≤0.10) and 199 downregulated (fold-change ≤−1.5 and adjusted $p$ value ≤0.10)[69–71]. GSEA was performed to examine pathway enrichment for the

differential expressed genes with the Molecular Signature Database "Hallmarks" gene set collection[44]. Only those with an FDR ≤0.25 were selected. MA-plot was generated using R ggpubr library, while heatmaps were generated using Multi-Experiment Viewer tool[72]. The RNAseq raw data are publicly available in the ArrayExpress repository under accession number: E-MTAB-11137.

**Metabolic assays**. The real-time oxygen consumption rate (OCR) of human immortalized glioblastoma cells (U87MG) was measured at 37 °C using a Seahorse XF Analyzer (Seahorse Bioscience, North Billerica, MA, USA). U87MG cells were transfected with praja2 siRNA, with praja2 and KSR2 siRNAs or with a control, scrambled siRNA (siRNAc). Forty-eight hours after transfection, cells were plated into specific cell culture microplates (Agilent, USA) at the concentration of $3 \times 10^4$ cells/well, and cultured for the last 12 h in DMEM, 10% FBS. OCR was measured in XF media (non-buffered DMEM medium, containing 10 mM glucose, 2 mM L-glutamine, and 1 mM sodium pyruvate) under basal conditions and after the sequential addition of 1.5 μM oligomycin, 2 μM FCCP, and rotenone + antimycin (0.5 μM all) (all from Agilent). Indices of mitochondrial respiratory function were calculated from the OCR profile: basal OCR (before the addition of oligomycin), basal OCR, maximal respiration (calculated as the difference between FCCP rate and antimycin+rotenone rate), spare respiratory capacity (calculated as the difference of FCCP-induced OCR and basal OCR), ATP production (calculated as the difference between basal OCR and oligomycin-induced OCR). Reported data were the mean values ± SEM of four measurements deriving from four independent experiments. The acidification rate (ECAR) of human immortalized glioblastoma cells, U87 cells, was measured at 37 °C using a Seahorse XF Analyzer (Seahorse Bioscience, North Billerica, MA, USA). U87 were transfected with praja2 siRNA, with both praja2 and KSR2 siRNAs, and with a scrambled siRNA. 48 hours after transfection, $3 \times 10^4$ cells were reseeded in triplicate into specific cell culture microplates (Agilent, USA), previously coated with polylysine. ECAR was measured in glucose-deprived XF media (non-buffered DMEM medium, 2 mM L-glutamine, and 1 mM sodium pyruvate) under basal conditions and after the sequential addition of Glucose (10 μM), Olygomycin (1 μM), and 2 DG (50 mM).

**Preparation of hybrid self-assembling nanoparticles (SANPs) encapsulating siRNA**. SANPs were prepared as reported previously[45]. Thus, calcium-phosphate colloidal dispersion (CaP NPs) was prepared. Briefly, an aqueous solution of dibasic hydrogen phosphate (10.8 mM, pH 9.5) was added 1:1 v/v, drop by drop and under magnetic stirring, to an aqueous solution of calcium chloride (18 mM, pH 9.5) for 10 min, and filtered with RC filter (0.22 μm filter membranes of regenerated cellulose). CaP NPs were prepared before the use, and then mixed with an aqueous solution of siRNA scramble (Sc) or praja2 by vortex for 10 sec in a ratio of 8:1 v/v, followed by incubation for 10 min, finally resulting in CaP/siRNA NPs. Different PEGylated cationic liposomes (PLs) consisting of DOTAP/chol/DSPE-PEG2000 (mM ratio 1:1.8:0.125), DOTAP/DSPE-PEG2000 (mM ratio 1:0.125), DC-chol/DSPE-PEG2000 (mM ratio 1:0.125), DC-chol/CER-PEG (1:0.125 mM), DOTAP/chol/CER-PEG (mM ratio 1:1.8:0.125), and DOTAP/CER-PEG (mM ratio 1:0.125) were prepared by hydration of a thin lipid film followed by extrusion (Supplementary Table 3). In the case of formulations prepared with the fluorescent marker, Rhod (1% w/w of total lipids) was added to the lipid mixture. The lipid film was obtained by a rotary evaporator (Laborota 4010 digital, Heidolph, Schwabach, Germany) and hydrated with RNAse-free water, for 2 h. Then, liposome suspension was extruded by a thermobarrel extruder system (Northern Lipids Inc., Vancouver, BC, Canada) passing repeatedly the suspension under nitrogen through polycarbonate membranes with decreasing pore sizes from 400 to 100 nm (Nucleopore Track Membrane 25 mm, Whatman, Brentford, UK). Finally, for transferrin-targeted PLs, Tf was firstly thiolated using 2-iminothiolane (Traut's reagent). Briefly, Tf was dissolved in 0.1 M Na-borate buffer pH 8, followed by the addition of Traut's reagent (1:5 mol/mol). Thereafter, thiolated Tf was incubated with preformed PLs DOTAP/CER-PEG/DSPE-PEG2000-Mal (CER-PEG/DSPE-PEG2000-Mal 95:5 w/w), overnight, at room temperature. The unconjugated Tf was removed by molecular exclusion chromatography, using a Sepharose G-25 column. PLs were stored at 4 °C. Finally, SANPs-siRNA were prepared by mixing, in equal volume, CaP/siRNA NPs and different PLs by vortex, for 10 s. Plain SANPs, without siRNA, were prepared similarly. Each formulation was prepared in triplicate.

**SANPs-siRNA characterization**. Plain SANPs (without siRNA) and SANPs-siRNA were characterized in terms of mean diameter, polydispersity index (PI), and zeta potential (ζ). Measurements were performed by dynamic light scattering after sample dilution by the Nanosizer Ultra (Malvern, UK). For each formulation, the mean diameter, the PI, and the ζ values were calculated as the mean of measurements carried out on at least three different batches. Moreover, in the case of SANPs-siRNA, the siRNA encapsulation efficiency was also determined in the different formulations by an indirect measure of unencapsulated siRNA, separated by ultracentrifugation (Optima Max E, Beckman Coulter, USA) at 80,000 rpm, 4 °C, for 40 min. The supernatants were analyzed by UV (UV-1800, UV Spectrophotometer) at the wavelengths of 260 nm, and the concentration of siRNA was calculated by a calibration curve of siRNAc and praja2 ($R^2 = 0.999$) in $H_2O$. Each analysis was carried out in duplicate.

**Animals**. CD1 4–6 weeks old male nude mice (20–22-g body weight; Charles River, Calco, CO, Italy) were kept under controlled conditions (temperature, 22 °C; humidity, 40%) on a 12-h light/dark cycle with food and water ad libitum. Mice received standard rodent pelleted chow 4RF21 (Mucedola, Settimo Milanese, Italy) and water ad libitum. All animal experiments were performed in accordance with guidelines approved by the Italian Ministry of Health (589/2017-PR; July 21, 2017).

**Orthotopic models of glioblastoma and SANPs treatments**. U87MG cells were stereotaxically implanted into the left caudate nucleus (by using the following coordinates: 0.6 mm anterior to the bregma; 1.7 mm lateral to the midline; and 3.5 mm ventral from the surface of the skull of male mice under ketamine (100 mg/kg, i.p.)/xylazine (10 mg/kg, i.p.) anesthesia[29]. Cells ($0.5 \times 10^6$ cells/5 μl) were implanted at an infusion rate of 1 μl/min. The needle was left in place 5 min after cell infusion before it was withdrawn. After 1 week from implantation, treatments with SANPs-siRNAs particles were performed as follows. Mice were treated every 48 h for 3 weeks by tail vein injection with SANPs-siRNAc (rhodamine-conjugated nanoparticles encapsulating siRNA scramble, control group) or SANPs-siPraja2 (rhodamine-conjugated nanoparticles encapsulating siRNA for praja2, treated group). At the end of treatment, mice were sacrificed and isolated brains were frozen in isopentane at −80 °C until the inclusion in OCT. For each brain, 10-μm-thick serial sections, from the beginning of the striatum to the hippocampus (1 section every 400 μm), were sliced. Subsequently, the sections stained with Mayer's hematoxylin and eosin (both from Diapath, Bergamo, Italy) and subjected to analysis for the quantification of tumor volume. The volumetric analysis was performed using software that measured the tumor area in each section and calculated the volume of the tumor according to Cavalieri's method using the following formula: $V = \sum(A)_i \times TS \times n$, where $(A)_I$ is the area of the tumor in level i, TS is the section thickness, and $n$ is the number of sections disposed between the two levels[73]. Where indicated, the mouse model of glioblastoma was generated by intracerebral injection of $3 \times 10^5$ U87 MG-Luc cells in 5 μL of saline solution (NaCl 0.9%) in 4–6 weeks old CD1 mice deeply anesthetized. To verify the successful injection, bioluminescent intensity (BLI) was measured after 3 h with IVIS Spectrum (Perkin-Elmer, Waltham, MA, USA), after intraperitoneal injection of 150 mg/kg D-Luciferin potassium salt (Perkin-Elmer). According to BLI measurement after 7 days, mice were randomized into three experimental groups of 12 animals, and each group was treated with SANPs-siRNAc or SANPs-siPraja2 by tail vein injection. Treatments were repeated twice a week for 4 weeks; then four mice for each group were sacrificed and organs were collected. Eight animals for each group were monitored until they showed a weight loss between 10 and 20%.

**Collection of human glial tumors**. The immunohistochemical analysis for praja2 was carried out on low-grade and high-grade glioma samples (from Neurosurgery patients of IRCCS Neuromed, Pozzilli, Italy) classified according to histopathological WHO 2016 classification of CNS tumors. A total of 20 cases of glial tumors were used: ten gliomas carrying wild-type IDH1 and ten gliomas with IDH1-R132 mutation. The material used in this study represents waste material and all patients have given their informed consent.

**Immunohistochemistry and biodistribution of rhodamine-conjugated SANPs-siRNA in brain sections**. Brain OCT embedded sections were fixed with 70% ethanol and immunostained for the proliferation marker Ki67 (Roche prediluted), praja2 (1:100, Bethyl), MOB1 (1:50, Cell signaling), and KSR2 (Santa Cruz Biotech) with a Benchmark Ultra XT (Roche). Antibody detection was performed by using ultraview DAB detection kit (Roche). Negative control samples were incubated with secondary antibodies only. The number of Ki67-positive and total cells was determined in five random 100x magnification fields by using Zeiss Epifluorescence Microscope Nikon Eclipse 50i equipped with a Nikon DS-Ri1 camera. To demonstrate the internalization of the rhodamine-conjugated nanoparticles, a separate experiment was carried out. The nude mice inoculated stereotaxically were treated for 3, 6, 9, and 16 h with rhodamine-conjugated nanoparticles. At the end of the treatment, the animals were sacrificed and the fresh brains were included in OCT. The OCT embedded sections were fixed in 70% ethanol and nuclei counterstained with DAPI (VECTASHIELD Antifade Mounting Medium for Fluorescence) were examined under a fluorescence microscope (Zeiss Epifluorescence Microscope Nikon Eclipse 50i). For praja2 staining, formalin-fixed, paraffin-embedded tissues from the tumors were selected. Representative slides of each tumor were stained with hematoxylin and eosin. Immunohistochemistry for praja2 was performed automatically with a Nexes instrument (Ventana, Tucson, Ariz.). Antibody detection was performed using a multilink streptavidin–biotin complex method, and antibodies were visualized by a diaminobenzidine chromagen method. Negative control samples were incubated with primary antibodies only. The number of praja2-positive cells was determined in fifty random fields ($75 \times 100$ μm$^2$ each) by using the Image Pro Plus 6.2 software and a stereological microscope Zaiss Image M11. Our aim was to obtain objective baseline data for the study of the expression of praja2 in these tissues. The results were expressed in numerical densities of positive cells in low-grade glioma and high-grade tissue and positive cells for fields at 100x magnification.

**Statistics and reproducibility**. Each experiment was repeated three to five times as described in the legend figures. All results are expressed as mean ± SEM in dot plots. Data distribution and gene expression statistical analyses were performed using GraphPad Prism software (v5.0; GraphPad Software Inc., San Diego, CA), Microsoft Excel 2016 (v16.04471; Microsoft Office 2016) and Interactive Dotplot (http://statistika.mfub.bg.ac.rs/interactive-dotplot/). Comparisons of two groups were performed using a Student's t-test. A p value of <0.05 was considered to be statistically significant.

**Reporting summary**. Further information on research design is available in the Nature Research Reporting Summary linked to this article.

## Data availability

The RNAseq raw data are publicly available in the ArrayExpress repository under accession number: E-MTAB-11137. Proteomic data generated in this study are available within the article and its supplementary data files. Other data used for the network analysis were extracted from publicly available databases: IntAct Molecular Interaction Database and CPTAC-GBM discovery cohort protein assay data. The mass spectrometry proteomics data have been deposited to the ProteomeXchange Consortium via the PRIDE [1] partner repository with the dataset identifierPXD033734. Uncropped images of blots/gels, Source Data for Figs. 1–6 and Supplementary Figs. 1–6 are available in Supplementary Data 2. All other relevant data are available from the corresponding author on reasonable request.

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

## Acknowledgements

This work was supported by Associazione Italiana per la Ricerca sul Cancro (AIRC, grants IG2018-ID22062 and IG-23068), the Italian Ministry of University and Research (PON 2018, grant PerMedNet, CUP: D26C18000260005), the European Regional Development Fund (POR Campania FESR 2014/2020, grants RarePlatNet, SATIN, Az. 1.2, CUP: B63D18000380007 and GENOMAeSALUTE, Az. 1.5, CUP: B41C17000080007) and Italian Ministry of Health (grant GR-2018-12366312, CUP: D19C20000650001) to A.F., A.W., G.G., and A.S. F.C. was supported by fellowships from the Italian Ministry of University and Research (grant PerMedNet) and PRIN2017 (grant 2017237P5x).

## Author contributions

R.I., Laura R., R.D.D., Luca R., M.A.O., E.S., D.B., L.L., S.C., F.C., C.G., and A.A. performed the experiments and analyzed the data. C.D.A. and Andrea S performed

proteomic experiments and analyzed the corresponding data. G.G., F.R., and Assunta S. performed the RNAseq gene network analysis; G.S. analysed the PPI subnetwork; G.D.R., V.C., and V.N. prepared the siRNA-SANPs. A.F., A.A., Andrea S., A.W., and C.A. conceived the experiments, supervised and analyzed the data, and revised the manuscript. A.F. wrote the manuscript.

## Competing interests

The authors declare no competing interests.
