## [Peer Review file · Communications Biology]

Reviewers' comments:

Reviewer #1 (Remarks to the Author):

In this manuscript, Donne et al investigate the UPS as a therapeutic target to reduce GBM growth. Whereas, their study globally supports their conclusion and bring some interesting therapeutic hypothesis, the authors draw overstated conclusions from their experiments. Furthermore, a lot of key experimental information and/or control experiments are missing in order to analyze/interpret the results in a rigorous and reproducible manner.

Thus, before considering for potential publication, some major issues have to be addressed.

Figure 2 Interaction praja2 with KRS2 in GBM

Supplementary experimental information AND experiments are definitively required to validate 1) the interaction between the Praja/KSR2 and between the 3 partners in a trimeric complex, 2) the impact of Praja on KSR2 stability. Importantly, there is only one experiment performed on GBM cells, the U87 cell line. Given the scope of the publication, main experiments should be also done in these cells.

2A : the authors presented all proteomic network, which might be misleading. The 3 analyses should be presented independently

2B : the authors presented a coimmunoprecipitation of different constructs of Flag-Praja2 using an anti-myc Ab. However, in their experiments, they are not presenting Flag-praja2 full length, only the dominant negative. This is a key experiment to perform and present. The authors should have also performed coIP using anti-Flag Ab and then realized WB using anti-KSR2-Myc.

2C : colocalization of KSR2 and Praja2. In regards to their coimmunoprecipitation, PLA experiments should have been performed to confirm interaction between the 2 proteins. I do not understand the purpose of the 4th square in the figure.

2D : the authors are stating the presence of a trimeric complex Praja-Myc-AMPKa1. First, they did not precised which antibody was used neither within the figure nor in the legend. Second, they performed coIP using only one antibody so they can only conclude that a significant portion of the targeted protein (Myc?) is able to interact with the other 2 partners. However, they do not have any proof that they are interacting at the same time, in a trimeric complex. Furthermore, the intensity of the band of both KSR2-Myc and AMPKa1 are very faint (in particular as compare to Fig2B).

2E and 2F : The experimental procedures for this assay is not presented in the material and method section. Did the authors immunoprecipitated or coIP the protein before performing WB? This is crucial in order to draw any conclusion.

2G : there is no time indication for the experiment. To confirm these results, experiments using proteasome inhibitors should have been performed. Another key experiment would have been to determine KSR2 half-life following inhibition of its translation.

Figure 3 Praja2 restrains AMPKa1 signaling and support the glycolytic pathway

Several key experiments are missing in this part, in particular control experiments, in order to draw significant and robust conclusions (see comments below). Furthermore, the authors focused their conclusion on glycolysis whereas significant metabolic differences are observed for OCR and only weak or no differences are observed for ECAR.

The authors should have performed all their si experiments using at least 2 different si sequences. How did si-praja2 interfere with KSR2 expression? If their previous conclusions are correct, an increased expression of KSR2 should be observed after si-KSR2.

The authors described a glycolytic metabolism of U87 cells. However, how did they reach this conclusion is not clear.

What are the impacts of si-KSR2 alone and AMPKa1 inhibitor on OCR and ECAR? (per lysate, which does not mean anything, or per cells?).

How si-praja or si-KSR2 impact substrate dependency or metabolic fueling?

How the si-KSR2 impact AMPKa1 phosphorylation?

What is the link between KSR2/AMPKa1 (stability? Ub? Location?)

What is the expression and phosphorylation of AMPKa1 after combination of praja and KSR2 silencing?

What does glucose deprivation impact cell metabolism when combined with praja or KSR2 silencing?

Usually, spare capacity and ATP-linked OCR are expressed as a % of initial OCR (3F).

The x-axis and the y-axis legends have to be verified as well as the normalization methods

Figure 5 Transcriptional reprogramming of praja-silenced cells

The direct impact of si-praja 2 (and KSR2) on cell cycle and cell proliferation should be determined and compared to SANPs-si praja. The effect of si-praja on cell proliferation is surprising given 1) snap-si praja efficacy (5C) and 2) cell cycle analysis (5E-F). How many different cultures of SANP-si Praja were performed in 5D (technical or experimental replicates?). Cell cycle analyses should be performed using classical protocols (Ip alone or BrDU/IP).

Figure 6 systemic delivery of SANP-si praja in vivo

Is it a perfusion or an i.v. injection of SANP? iv injection every 48 hours for how long? Why are the mice sacrificed at 3 weeks, how did the authors choose this time-point? (maybe Fig7 should be Figure 6)

The authors should performed an immunostaining for tumor cells (MHC I for example) in order to show that SANP are indeed within the tumor (6B). IHC staining images are in general of poor quality. That should be improved.

It would have been better and more informative if the authors presented a dot plot rather than a histogram for Fig 6D.

Why did the authors look at MOB1 expression, what is the relevance with praja/KSR2? What about AMPKa1 phosphorylation?

Reviewer #2 (Remarks to the Author):

In this study, Delle Donne et al reported that IDH1 wildtype GBM preferentially expressed Praja2, a RING E3 ubiquitin ligase. By ubiquitinating and degrading KSR2, Praja2 indirectly controls AMPK activity and its associated cellular metabolism. Finally, the authors showed that the delivery of siRNA targeting Praja2 by using transferrin-targeted self-assembling nanoparticles (SANP) prevented KSR2 degradation, inhibited GBM growth and prolonged the survival rate of treated mice. The link between ubiquitination and GBM metabolism mediated by Praja2 is interesting. It is also commendable that the authors attempted to translate their findings into nanoparticle-mediated GBM therapies. However, the current manuscript suffers from several shortcomings which must be addressed before the manuscript can be accepted.

Major points:

1) The authors used only one siRNA against Praja2 in the entire paper. Given that siRNAs are established to have off-target effects, it is important for the authors to repeat some key experiments with another Praja2 siRNA, or to perform a rescue experiment by expressing Praja2 in the setting of Praja2 knockdown (KD).

2) The authors showed that IDH1 wildtype but not IDH1 mutant GBM preferentially expressed Praja2. Is this linked to the distinct metabolic state between IDH1 wildtype vs mutant GBM that is mediated by Praja2? Also, how many IDH1 wildtype and IDH1 mutant GBM tissues were used to quantify Praja2+ cells in figure 1B?

3) In figure 2 (e.g. E and F), the effect of Praja2 overexpression or KD on KSR2 protein levels is modest and inconsistent. To strengthen their point, the authors need to identify the ubiquitination site of KSR2 and show that Praja2 can ubiquitinate KSR2 in vitro. Can the KSR2 degradation by Praja2 be blocked by proteasome inhibition? It is also unclear if the Praja2RM mutant is indeed catalytically inactive since the authors did not indicate how Praja2RM was constructed.

4) In figure 4, the authors showed that Praja2 KD upregulated several pathways, including OXPHOS, and that this was associated with increased activity of PPARGC1A, PPARGC1B and NRF1. Given that Praja2 is not a transcription factor, the authors should better explain how Praja2 can affect gene expression in GBM cells.

5) In figure 7C, the survival curve showed that some SANPs-siPraja2 treated mice were still alive for as long as 70 days post-implantation. The authors need to show the bioluminescence images of all the mice in this experiment as this does not corroborate with the data in figure 7A-B. Also, is the variation in the survival of SANPs-siPraja2 treated mice due to different Praja2 KD efficiency?

Reviewer #3 (Remarks to the Author):

Donne et al identify praja2 as a regulator of cancer cell metabolism and a potential therapeutic target to suppress GBM growth.

I have several questions regarding the manuscript:

1. Is praja2 also transcriptionally expressed at higher levels in IDH wt GBM? Instead of IDH mut GBM, the authors can classify these tumors as IDH mut astrocytoma as per new CNS tumor classifications (assuming these are astrocytoma). More detail can be provided about the number of samples, number of patients etc for Fig. 1.

2. the adjusted p-value (0.1) for the analysis described in Fig. 4 is quite high. How many replicates were used for RNA-seq?

3. Are the SANPs-siRNAs perfused through the caudal vein 2x or 3x per week? The mice used are female or male? Some clarifications are needed.

Reviewer #1 (Remarks to the Author):

In this manuscript, Donne et al investigate the UPS as a therapeutic target to reduce GBM growth. Whereas, their study globally supports their conclusion and bring some interesting therapeutic hypothesis, the authors draw overstated conclusions from their experiments. Furthermore, a lot of key experimental information and/or control experiments are missing in order to analyze/interpret the results in a rigorous and reproducible manner.

Thus, before considering for potential publication, some major issues have to be addressed.

R. *We wish to thank the Reviewer to find our study interesting and globally supported by the findings. We agree with the Reviewer that conclusions of some experiments are overstated. We also apologize that some of the key experimental information and control experiments were missing. We carefully followed Reviewer's suggestions repeating key experiments, strengthening the relevance of the findings and providing all the missing information. Please, see below for the changes made.*

Figure 2 Interaction praja2 with KRS2 in GBM

Supplementary experimental information AND experiments are definitively required to validate 1) the interaction between the Praja/KSR2 and between the 3 partners in a trimeric complex, 2) the impact of Praja on KSR2 stability. Importantly, there is only one experiment performed on GBM cells, the U87 cell line. Given the scope of the publication, main experiments should be also done in these cells.

R. *As suggested, we performed reverse co-immunoprecipitation experiments in GBM cells between praja2, KSR2 and AMPKa using anti-Myc antibody for IP. The data shown in the **new Supplementary Fig .2A** indicate that praja2, KSR2 and AMPKa form a complex in GBM lysates. Moreover, we demonstrated that in GBM cells praja2 promotes KSR2 degradation through the proteasome (please, see **new Fig.2J and 2K**, page 6, lines 146-148 and 165-167).*

2A: the authors presented all proteomic network, which might be misleading. The 3 analyses should be presented independently

R. *According to the Reviewer's suggestion, we splitted the network in three independent subnetworks (**new Supplementary Fig. 1B**). However, we think that the main message to the reader might be not clear. For this reason, we highlighted the connecting lines between praja2 and partners identified in different studies and have included a modified version of the network also in the main figure (**new Fig. 2A**, page 5, lines 128-132).*

2B : the authors presented a coimmunoprecipitation of different constructs of Flag-Praja2 using an anti-myc Ab. However, in their experiments, they are not presenting Flag-praja2 full length, only the dominant negative. This is a key experiment to perform and present. The authors should have also performed coIP using anti-Flag Ab and then realized WB using anti-KSR2-Myc.

R. *We wish to note that co-expression of wild type praja2 with KSR2 leads to proteolysis of the bound kinase. To prevent this, we should use the proteasome inhibitor MG132. However, the use of MG132 for longer time is quite toxic for cells and often results in cell death. To avoid MG132 treatment, we prefer to use the RING inactive mutant of the ligase for co-IP experiments of exogenous proteins. Nevertheless, to accomplish the Reviewer's suggestion, we repeated the triple co-IP experiments using praja2-flag and reverse coimmunoprecipitation with anti-myc ab for IP, in the presence of MG132 (the treatment was performed for 6 hours). The results shown in the **new Supplementary Fig.2A** confirmed the formation of a trimeric complex between praja2, KSR2 and AMPK. (Page 6, lines 146-148).*

2C: colocalization of KSR2 and Praja2. In regards to their coimmunoprecipitation, PLA experiments should have been performed to confirm interaction between the 2 proteins. I do not understand the purpose of the 4th square in the figure.

R. *To demonstrate the physical interaction between praja2 and KSR2, we performed in vitro pull-down assays using in vitro synthesized, recombinant proteins. The results demonstrate that praja2 directly interacts with KSR2 (please, see **new Fig. 2C**). Moreover, we made the appropriate correction to the 4th square of **Fig. 2D** (page 6, lines 145-146).*

2D : the authors are stating the presence of a trimeric complex Praja-Myc-AMPKa1. First, they did not precised which antibody was used neither within the figure nor in the legend. Second, they performed coIP using only one antibody so they can only conclude that a significant portion of the targeted protein (Myc?) is able to interact with the other 2 partners. However, they do not have any proof that they are interacting at the same time, in a trimeric complex. Furthermore, the intensity of the band of both KSR2-Myc and AMPKa1 are very faint (in particular as compare to Fig2B).

R. *We apologize for the missing information about which antibody was used for Co-IP experiments. We have now added this information in the **Fig. 2E** and **legend** (page 15, line 450-452). Moreover, we performed reverse co-immunoprecipitation experiments between praja2, KSR2 and AMPKa using, at this time, anti-Myc antibody for IP. As indicated above, the data shown in the **new Supplementary Fig.2A** confirmed the formation of a trimeric complex between praja2, KSR2 and AMPK. As for the intensity of bands found in the IP, this may depend on the dynamic nature of the interaction, stoichiometry of co-expressed proteins, efficiency of immunoprecipitation, ECL exposure, and more. We have explained in the text that only a portion of AMPKa and KSR2 are, indeed, in the complex with praja2 (Page 6, lines 146-148).*

2E and 2F : The experimental procedures for this assay is not presented in the material and method section. Did the authors immunoprecipitated or coIP the protein before performing WB? This is crucial in order to draw any conclusion.

R. *Immunoblot analysis was performed on KSR2 immunoprecipitates (**new Fig. 2F and 2G**). Apologies for the missing information. We have now added this information in the manuscript (page 15-16, lines 452-458).*

2G : there is no time indication for the experiment. To confirm these results, experiments using proteasome inhibitors should have been performed.

R. *We repeated experiments using the proteasome inhibitor MG132. The data show that treatment with MG132 reversed, at least in part, the effects of praja2 on KSR2 levels, indicating that KSR2 degradation occurs through the proteasome (please, see **new Fig. 2J and 2K**). We have also indicated the time points in the legend (page 16, lines 461-463).*

Another key experiment would have been to determine KSR2 half-life following inhibition of its translation.

R. *As suggested by the Reviewer, we performed experiments aimed to define the KSR2 half-life in the presence or absence of praja2 with the protein translation inhibitor cycloheximide. As shown in the **new Supplementary Fig. 2C**, praja2 expression accelerates KSR2 proteolysis (page 6, lines 164-165). Interestingly, we found that also praja2 levels decay overtime from cycloheximide treatment, indicating that the half-life of the ligase is quite low. We will further pursue this interesting aspect in a different work.*

Figure 3 Praja2 restrains AMPKa1 signaling and support the glycolytic pathway
Several key experiments are missing in this part, in particular control experiments, in order to draw significant and robust conclusions (see comments below). Furthermore, the authors focused their conclusion on glycolysis whereas significant metabolic differences are observed for OCR and only weak or no differences are observed for ECAR.

R. *We wish to thank the Reviewer for this important comment that helped us to perform new experiments including the appropriate controls and address the main criticisms. Please, see below.*

The authors should have performed all their si experiments using at least 2 different si sequences.

R. We have now included the information concerning the use of siRNAs for praja2 in the revised manuscript (page 20, lines 593-599). In particular, in this manuscript and previous studies (Lignitto et al., *Nature Comm.* 2013, 4:1822; Sepe et al., *PNAS* 2014, 111:15729), we used a pool of 4 siRNAs targeting praja2. Potential off-target effects of siRNAs have been excluded by performing rescue experiments or using single siRNA preparations. Nevertheless, following the Reviewer's comment, we repeated growth curve experiments in GBM cells using two different praja2 siRNAs and found comparable effects to those observed with a pool of all 4 siRNAs (Please, see **the new Supplementary Fig. 5**) (Page 9, lines 271-274).

How did si-praja2 interfere with KSR2 expression? If their previous conclusions are correct, an increased expression of KSR2 should be observed after si-KSR2.

R. We have addressed this issue by analyzing KSR2 levels in praja2-silenced cells. As expected, downregulation of praja2 significantly increased the basal levels of KSR2 (**new Fig. 3E and Supplementary Fig. 3**) (page 7, lines 184-185 and page 6, lines 167-169).

The authors described a glycolytic metabolism of U87 cells. However, how did they reach this conclusion is not clear. What are the impacts of si-KSR2 alone and AMPK α 1 inhibitor on OCR and ECAR? (per lysate, which does not mean anything, or per cells?). How si-praja or si-KSR2 impact substrate dependency or metabolic fueling?

R. As suggested, we repeated metabolic analyses measuring both OCR and ECAR. show that praja2 silencing markedly enhanced OCR with minor effects on ECAR, indicating that the ligase mostly controls respiratory chain activity (**New Fig 3G,J and Supplementary Fig. 4A and 4D**). Furthermore, we repeated the metabolic analysis in GBM cells transfected with siRNA targeting KSR2 or pretreated with the AMPK inhibitor (SBI-0206965). As shown in the **new Fig. 3G, J**, KSR2 silencing or AMPKi treatment had no major impact on OCR (please, see **new Fig. 3G,J**). We have now better explained these aspects in the revised manuscript (page 7, lines 186-210).

How the si-KSR2 impact AMPK α 1 phosphorylation? What is the link between KSR2/AMPK α 1 (stability? Ub? Location?)

R. Previous work by others demonstrated that KSR2 directly interacts with AMPK α 1, favoring its activation by AMP. By supporting AMPK activation, KSR2 promotes glucose and fatty acid catabolic pathways. Accordingly, mice bearing KSR2 deletion are hypothermic, hypophagic and develop obesity (**Cell Metabolism, 2009,10:366-78**). Mechanistically, it is still not clear how KSR2 regulates AMPK activity. As also reported by others, we do not see major effects of KSR2 on AMPK stability. Most likely, KSR2 works as a molecular platform for the assembly of multimeric complexes including mitogenic (MEK/ERK) and metabolic (AMPK, Lkb1) kinases, and perhaps phosphatases, facilitating up- and downstream activation of proliferative and metabolic pathways. Further substantial work is required to better address this important issue.

What is the expression and phosphorylation of AMPK α 1 after combination of praja and KSR2 silencing?

R. As suggested, we have now analyzed the phosphorylation and expression of AMPK in KSR2- and praja2-silenced cells. The results indicate that concomitant downregulation of KSR2 reversed, at least in part, the effects of praja2 silencing on AMPK (**new Fig.3E and 3F**, page 7, lines 184-185).

What does glucose deprivation impact cell metabolism when combined with praja or KSR2 silencing?

R. To address this point, we performed metabolic assays using GBM cells incubated with DMEM without glucose supplementation. Before the assay, cells were treated with glucose and then with 2-

deoxyglucose. As shown in the **new Supplementary Fig. 4E and 4F**, genetic silencing of praja2 and KSR2 minimally impacted on ECAR, confirming a more prominent role of praja2-KSR2 axis in the control of oxidative metabolism (page 7, lines 202-203).

Usually, spare capacity and ATP-linked OCR are expressed as a % of initial OCR (3F).

The x-axis and the y-axis legends have to be verified as well as the normalization methods

R. We apologize for the confusion. We have now corrected legend to **new Fig.3H** (page 16, lines 483-487) and Materials & Methods (page 24, line 749-761). Moreover, we have included a supplementary panel showing the spare capacity expressed as % (**new Supplementary Fig. 4B and 4D**, page 7, lines 194-205).

Figure 5 Transcriptional reprogramming of praja-silenced cells

The direct impact of si-praja 2 (and KSR2) on cell cycle and cell proliferation should be determined and compared to SANPs-si praja. The effect of si-praja on cell proliferation is surprising given 1) snap-si praja efficacy (5C) and 2) cell cycle analysis (5E-F). How many different cultures of SANP-si Praja were performed in 5D (technical or experimental replicates?). Cell cycle analyses should be performed using classical protocols (Ip alone or BrDU/IP).

R. As suggested, we comparatively evaluated the impact of si-praja2 and SANPs-si praja2 treatment on GBM cell growth by repeating growth curves and FACS analyses. Cumulative data are now shown in the **new Fig. 5F** and are expressed as mean value \pm SEM of five independent experiments, while FACS analyses show representative results of three independent experiments. We have added this information in the figure legend (page 9, lines 276-278 and page 18, lines 525-526).

Figure 6 systemic delivery of SANP-si praja in vivo. Is it a perfusion or an i.v. injection of SANP? iv injection every 48 hours for how long? Why are the mice sacrificed at 3 weeks, how did the authors choose this time-point? (may be Fig7 should be Figure 6)

R. One-week post-implantation, SANPs-siRNAs were i.v. injected into the caudal vein every 48h, for a total of 14 days of treatment. At three weeks post-implantation, mice were sacrificed and tumor lesions isolated and further characterized. We have corrected the schematic panel and the legend to **new Fig. 6A** (page 18, lines 529-533). The orthotopic model of GBM runs for 3-4 weeks, depending on the mice line. For CD133 strain used in this study, we were authorized to sacrifice the mice at three weeks post-implantation to avoid animal suffering. During the experiment, the cut-off was established from time to time according to the criteria indicated in the **Table below**. In addition to the terms defined according to the procedures described above, there are limits of severity (human endpoints) based on the clinical status of the animal and signs of suffering. The attain of these limits led to the exclusion of the animal from the experimental procedure and in some cases, after evaluation by the designated veterinarian, the immediate sacrifice of the animals (page 10, lines 280-287 and page 11, lines 314-318).

Table for Reviewer only.

CLINICAL EVENT	PAIN SCORE	ACTIONS
Normal coat, normal locomotion and general activity, normal consumption of water and food	0	NORMAL
Hair matting (pilo erection, injuries, dehydration)	1	DAILY MONITORING (1-3)
Hair matting, decreased physical activity (grooming, exploration)	2	
Hair matting, decreased physical activity (lethargy) weight loss <10%	3	
Hair matting, lethargy, weight loss <10%, breathing difficulties (dyspnoea / tachypnea)	4	

Hair matting, lethargy, weight loss > 10% and <20%, dyspnoea / tachypnea, abnormal postures (lordosis / kyphosis)	5	ANIMAL WITH INITIAL SIGNS OF SUFFERING (4-5)
---	---	--

The authors should have performed immunostaining for tumor cells (MHC I for example) in order to show that SANP are indeed within the tumor (6B). IHC staining images are in general of poor quality. That should be improved.

R. *As suggested, we repeated immunostaining experiments on mouse brain tumor sections using anti-vimentin antibody to better identify the implanted tumor (new Fig. 6B, page 10, lines 288-292). As requested, we have also provided high-quality images of all IHC (please, see new Fig. 1A, 6B, 6E, 6J).*

It would have been better and more informative if the authors presented a dot plot rather than a histogram for Fig 6D.

R. *We have now converted the histogram in dot plot analysis (Please, see the new Fig. 6D, page 10, lines 297-299).*

Why did the authors look at MOB1 expression, what is the relevance with praja/KSR2? What about AMPKa1 phosphorylation?

R. *Since praja2 ubiquitylates and degrades MOB1 (Nat. Comm. 2013, 4, 1822), downregulation of praja2 is expected to elevate MOB1 levels. Therefore, MOB1 staining in siRNA-treated GBM tumors was used as a positive control for praja2 silencing. However, as requested, we have now included IHC images for p-AMPK and AMPKa, both in controls (SANPsiRNAc) and SANPsiPraja2-treated mice (new Fig. 6J, page 11, lines 320-322).*

Reviewer #2 (Remarks to the Author):

In this study, Delle Donne et al reported that IDH1 wildtype GBM preferentially expressed Praja2, a RING E3 ubiquitin ligase. By ubiquitinating and degrading KSR2, Praja2 indirectly controls AMPK activity and its associated cellular metabolism. Finally, the authors showed that the delivery of siRNA targeting Praja2 by using transferrin-targeted self-assembling nanoparticles (SANP) prevented KSR2 degradation, inhibited GBM growth and prolonged the survival rate of treated mice. The link between ubiquitination and GBM metabolism mediated by Praja2 is interesting. It is also commendable that the authors attempted to translate their findings into nanoparticle-mediated GBM therapies. However, the current manuscript suffers from several shortcomings which must be addressed before the manuscript can be accepted.

R. *We wish to thank the Reviewer in finding “the link between ubiquitination and GBM metabolism mediated by Praja2 interesting, and commendable that we attempted to translate our findings into nanoparticle-mediated GBM therapies”. We followed the Reviewer’s suggestions and repeated key experiments, strengthening the relevance of the findings and providing all the missing information. Please, see below for changes made.*

Major points:

1) The authors used only one siRNA against Praja2 in the entire paper. Given that siRNAs are established to have off-target effects, it is important for the authors to repeat some key experiments with another Praja2 siRNA, or to perform a rescue experiment by expressing Praja2 in the setting of Praja2 knockdown (KD).

R. *We have now included more details concerning the use of siRNAs for praja2 in the revised manuscript (page 20, lines 593-599). In particular, we used a pool of 4 siRNAs targeting praja2. Potential off-target effects of siRNAs have been excluded by performing rescue experiments or using single siRNA preparations in previous work (Lignitto et al., Nature Comm. 2013, Sepe et al., PNAS 2014). Nevertheless, following the Reviewer’s comment, we repeated growth curve experiments in GBM cells using two different praja2 siRNAs and found comparable effects to those observed with the pool of all 4 siRNAs (Please, see **new Supplementary Fig.5**, and page 9, lines 271-274).*

2) The authors showed that IDH1 wildtype but not IDH1 mutant GBM preferentially expressed Praja2. Is this linked to the distinct metabolic state between IDH1 wildtype vs mutant GBM that is mediated by Praja2?

R. *Mutations in isocitrate dehydrogenase 1 (IDH1) are frequent in several cancer entities, including diffuse glioma. Mutated IDH1 contributes to cancer development and progression by reshaping metabolic pathways and altering the epigenetic landscape of glial tumor cells (Raineri et al. 2018, Front Genet 9:493). In this context, glioma lesions carrying mutated IDH1 are less aggressive and more prone to therapeutic approaches. Our data indicate that praja2 overexpression marks selectively glioma lesions with wild-type IDH1, most likely reflecting a peculiar metabolic landscape of these subtypes of tumors. We updated the references on this topic (Trautwein et al. 2018 JCI Insight, 7).*

*Following the Reviewer’s suggestion,, we have evaluated praja2 mRNA expression profiles in a TCGA-GBM cohort. Samples were then stratified based on their IDH status (wild-type vs. mutant) and their histological subtype (astrocytoma, glioblastoma, oligoastrocytoma, oligodendroglioma). As indicated in the text, the mRNA analysis was performed on 167 samples of astrocytoma (51 wild-type and 116 mutant), 235 samples of glioblastoma (216 wild-type and 19 mutant), 113 samples of oligoastrocytoma (15 wild-type and 98 mutant) and 166 samples of oligodendroglioma (16 wild-type and 150 mutant). The data shown in the **new Supplementary Fig. 1A** suggest that praja2 levels may discriminate the mutational status of IDH1 (Page 5, lines 117-124).*

Also, how many IDH1 wildtype and IDH1 mutant GBM tissues were used to quantify Praja2+ cells in figure 1B?

R. *Apologies for the missing information. We used 10 different glioma tissues for each experimental group. We added this information in the legend to **Fig. 1B** and in Materials & Methods (page 27, lines 836-841 and page 15, lines 430-431).*

3) In figure 2 (e.g. E and F), the effect of Praja2 overexpression or KD on KSR2 protein levels is modest and inconsistent. To strengthen their point, the authors need to identify the ubiquitination site of KSR2 and show that Praja2 can ubiquitinate KSR2 in vitro.

R. *We wish to note that ubiquitylation experiments of KSR2 are performed in the presence of MG132 to prevent KSR2 proteolysis by coexpressed praja2 (please, see **Fig.2F, 2G and below** and page 15, lines 452-458).*

Can the KSR2 degradation by Praja2 be blocked by proteasome inhibition?

R. *We have now repeated the experiments using the proteasome inhibitor MG132. The data indicate that praja2 promotes KSR2 degradation through the proteasome (please, see **new Fig. 2J and 2K**, page 6, lines 165-167).*

It is also unclear if the Praja2RM mutant is indeed catalytically inactive since the authors did not indicate how Praja2RM was constructed.

R. *Apologies for the missing information. We have included the appropriate information and reference in the Material & Methods (please, see page 20, lines 591-592).*

4) In figure 4, the authors showed that Praja2 KD upregulated several pathways, including OXPHOS, and that this was associated with increased activity of PPARGC1A, PPARGC1B and NRF1. Given that Praja2 is not a transcription factor, the authors should better explain how Praja2 can affect gene expression in GBM cells.

R. *We agree with the Reviewer's comment. We have now discussed this interesting point in the text (Please, see page 13, lines 385-388).*

5) In figure 7C, the survival curve showed that some SANPs-siPRaja2 treated mice were still alive for as long as 70 days post-implantation. The authors need to show the bioluminescence images of all the mice in this experiment as this does not corroborate with the data in figure 7A-B. Also, is the variation in the survival of SANPs-siPRaja2 treated mice due to different Praja2 KD efficiency?

R. *As requested, we have now added the bioluminescence images of all the mice of the in vivo experiment (please, see **Supplementary Fig. 6**). We believe that the variable efficiency of praja2 silencing in vivo may explain some variation in the survival rate of SANPsiPraja2-treated mice compared to controls.*

Reviewer #3 (Remarks to the Author):

Donne et al identify praja2 as a regulator of cancer cell metabolism and a potential therapeutic target to suppress GBM growth.

R. *We wish to thank the Reviewer for the helpful suggestions that helped to improve the manuscript.*

I have several questions regarding the manuscript:

1. Is praja2 also transcriptionally expressed at higher levels in IDH wt GBM? Instead of IDH mut GBM, the authors can classify these tumors as IDH mut astrocytoma as per new CNS tumor classifications (assuming these are astrocytoma).

R. *We have now evaluated praja2 mRNA expression profiles in a TCGA-GBM cohort. Samples were then stratified based on their IDH status (wild-type vs. mutant) and their histological subtype (astrocytoma, glioblastoma, oligoastrocytoma, oligodendroglioma). As indicated in the text, the mRNA analysis was performed on 167 samples of astrocytoma (51 wild-type and 116 mutant), 235 samples of glioblastoma (216 wild-type and 19 mutant), 113 samples of oligoastrocytoma (15 wild-type and 98 mutant) and 166 samples of oligodendroglioma (16 wild-type and 150 mutant). The data shown in the **new Supplementary Fig. 1A** suggest that praja2 levels may discriminate the mutational status of IDH1 (page 5, lines 117-124).*

More detail can be provided about the number of samples, number of patients etc for Fig. 1.

R. *Apologies for the missing information. We used 10 different glioma tissues for each experimental group. We added this information in the legend of figure 1 (page 15, lines 430-431) and in **Materials & Methods** (page 27, lines 836-841).*

the adjusted p-value (0.1) for the analysis described in Fig. 4 is quite high. How many replicates were used for RNA-seq?

R. *RNA-Seq was performed using three biological replicates (Materials and Methods section, page 23). In addition, to build a solid set of expressed genes, a given RNA was considered expressed when detected by at least ≥ 10 raw reads. After differential expression analysis, before setting the adjusted p-value cutoff to 0.10, we plotted its distribution. The cutoff was chosen based on prior RNA-Seq experiments, which show a good sensitivity from the next-generation sequencing approach, but the optimal adjusted p-value also accounts for false positives due to normalization. Moreover, considering that it is the first time that the effect of praja2 has been studied in this context, we have chosen to use a less stringent adjusted p-value in order to have a more global view of its effect. Our choice was also supported by the fact that several studies have performed RNA-Seq differential expression analysis setting the adjusted p-value to 0.1. We have now included the appropriate references in the **Materials and Methods section** (Ehmsen et al., 2019, *Sci. Data*; Sanghi et al, 2021, *Nature Comm.*; Sarma et al., 2021, *Nature Comm.*) (page 24, lines 730-733).*

3. Are the SANPs-siRNAs perfused through the caudal vein 2x or 3x per week? The mice used are female or male? Some clarifications are needed.

R. *One-week post-implantation, SANPs-siRNAs were i.v. injected into the caudal vein of male mice every 48h, for a total of 14 days of treatment. At three weeks post-implantation, mice were sacrificed and tumor lesions isolated and further characterized. We have corrected the schematic panel and legend of **Fig. 6A** (page 18, lines 530-536).*

REVIEWERS' COMMENTS:

Reviewer #1 (Remarks to the Author):

the authors significantly improved their manuscript by providing substantial missing information and experimental results.

Reviewer #2 (Remarks to the Author):

The authors have sufficiently addressed my previous queries and I agree to publish it.

Reviewer #3 (Remarks to the Author):

The authors have addressed my concerns.